

# Performance Evaluation of MeteoTracker Mobile Sensor for Outdoor Applications

Francesco Barbano[1], Erika Brattich[1], Carlo Cintolesi[1], Abdul Ghafoor Nizamani[1], Silvana Di Sabatino[1], Massimo Milelli[2], Esther E. M. Peerlings[3], Sjoerd Polder[3], Gert-Jan Steeneveld[3], and Antonio Parodi[2]

[1]University of Bologna, Dept of Physics and Astronomy, via Irnerio 46, 40126 Bologna, Italy
[2]CIMA research Foundation, Via A. Magliotto, 2 17100 Savona, Italy
[3]Wageningen University, Meteorology and Air Quality Section, P.O. box 47, 6700 AA, Wageningen, The Netherlands

**Correspondence:** Francesco Barbano (francesco.barbano3@unibo.it)

**Abstract.** The morphological complexity of urban environments results in a high spatial and temporal variability of the urban microclimate. The consequent demand for highly-resolution atmospheric data remains a challenge for atmospheric research and operational application. The recent widespread availability and increasing adoption of low-cost mobile sensing offers the opportunity to integrate observations from conventional monitoring networks with microclimatic and air pollution data at a

finer spatial and temporal scale. So far, the relatively low quality of the measurements and outdoor performance compared to conventional instrumentation has discouraged the full deployment of mobile sensors for routine monitoring. The present study addresses the performance of a commercial mobile sensor, the MeteoTracker (IoTopon Srl), recently launched on the market to quantify the microclimatic characteristics of the outdoor environment. The sensor follows the philosophy of the Internet of Things technology, being low cost, having an automatic data flow via personal smartphones and online data sharing, supporting

user-friendly software, and having the potential to be deployed in large quantities. In this paper, the outdoor performance is evaluated through tests aimed at quantifying (i) the intra-sensor variability under similar atmospheric conditions and (ii) the outdoor accuracy compared to a reference weather station under sub-optimal (in fixed location) and optimal (mobile) sensor usage. Data-driven corrections are developed and successfully applied to improve the MeteoTracker data quality. In particular, a recursive method for the simultaneous improvement of relative humidity, dew point, and humidex index proves crucial for

increasing the data quality. The results mark an intra-sensor variability in the range of $\pm0.5°$C for air temperature and $\pm1.2\%$ for the corrected relative humidity, both within the declared sensor accuracy. The sensor captures the same atmospheric variability as the reference sensor during both fixed and mobile tests, showing positive biases (overestimation) for both variables. Through the mobile test, the outdoor accuracy is observed between $\pm0.3°$C to $\pm0.5°$C for air temperature, between $\pm3\%$ and $\pm5\%$ for the relative humidity, ranking the MeteoTracker in the real accuracy range of similar commercial sensors from the literature

and making it a valid solution for atmospheric monitoring.

## 1 Introduction

The coverage of the Earth's surface by atmospheric monitoring networks remains challenging, especially in remote locations, poor countries, and complex terrain. Among the last category, the urban environment requires long-term monitoring at high





spatial and temporal resolutions, as turbulence structures play a key role in inertial and thermal ventilation (Barbano et al.,
2020; Cintolesi et al., 2021). To fill the gap, classical urban observational networks are supported by spot-on intensive field
campaigns for intra-urban flow detailing and turbulence analysis. The recent development of low-cost sensors provides a
novel opportunity to integrate the existing monitoring networks with cheaper, yet reliable solutions. Nowadays, monitoring
protocols for fixed low-cost weather stations have adopted crowdsourcing approaches (Meier et al., 2017; Fenner et al., 2021)
to increase spatial coverage of urban areas, with the creation of community networks (Jiao et al., 2016) for environmental
monitoring. Contextually, the adoption of mobile sensors and smartphones is increasing, carrying the typical shortcomings of
novel approaches, such as the lack of protocols for mobile sensing, outdoor accuracy, and long-term reliability. Data quality
from mobile sensing will require suitable but transferable sensor calibration strategies (Xu et al., 2019; DeSouza et al., 2022)
as well as the development of accurate correction algorithms (Huang et al., 2023). The development of dedicated platforms
(e.g., Den Ouden et al. 2021) provides a virtual environment where quality-controlled mobile data can be safely stored and
shared.

Two major classes of sensors have been developed according to their application scopes, i.e., the study of microclimate and
air quality. As a short notice, air quality sensors mostly monitor regulated pollutants, such as particulates, nitrogen dioxide
and ozone, and/or greenhouse gases such as carbon dioxide (e.g., Johnson et al. 2016; Van den Bossche et al. 2016; Puri et al.
2020; Gómez-Suárez et al. 2022; Ganji et al. 2023). Microclimate sensors are firstly designed for monitoring the urban thermal
environment (Kousis et al., 2022), but also the radiative properties of the atmosphere (Heusinkveld et al., 2023), evapotran-
spiration (Markwitz and Siebicke, 2019) and wind-related quantities (Droste et al., 2020) have found recent interest. A key
ensemble of this second category consists of mobile sensors suitable for measuring the thermo-hygrometric characteristics of
the atmosphere on the move. The sensor suite is typically composed of a thermo-hygrometer or a thermo-logger, but examples
of complete weather stations mounted on moving vehicles (Heusinkveld et al., 2014; Emery et al., 2021) or integration with
automatic infrared cameras (Lindberg, 2007; Acosta et al., 2022) are also documented. Two main categories of sensors are also
used: research-grade instrumentation designed for conventional weather stations and adapted for mobile use, and low-cost mo-
bile sensors. In addition, mobile sensing using smartphones is also accessed nowadays, thanks to the presence of temperature
sensors oftentimes installed within some devices (e.g., Cabrera et al. 2021) or the use of alternative data proxies such as the
temperature of the smartphone battery (Overeem et al., 2013; Droste et al., 2017), and the potential crowdsourcing from large
communities. oOftentimes, the sensor fabric is modified, adding homemade radiation shields (Sun et al., 2009; Leconte et al.,
2015) and ventilation pipes (Tsin et al., 2016). Despite walking being an explored option in the literature, the vast majority
of mobile monitoring is performed using bicycles or motor vehicles. This allows a wide spatial coverage of the urban and
surrounding areas (Sun et al., 2019), a large number of monitoring scans (Emery et al., 2021), a long-term assessment (Charabi
and Bakhit, 2011) and a variety of monitoring techniques, including spot-on measurements (Qaid et al., 2016) and transect's
inspections (Unger et al., 2001).

The top-trend research topic is the Urban Heat Island (UHI) effect, where mobile sensing offers a denser representation of the
canyon-level air temperature (Stewart, 2011) inside the urban context. Intra-urban UHI and local thermal effects are attributed
to land cover, urban morphology, and aspect ratio of the urban canyons. Yan et al. (2014) used an instrumented bicycle to infer



the magnitude and spatial characteristic of the air temperature variations related to the landscape parameters characterizing the immediate environment of the measurement sites. Focusing on the street canyon, Sun (2011) conducted a mobile survey by bike to show a positive correlation between the air temperature and the height-to-width ratio of the canyon, green area coverage, and building ratio. Covering a larger spatial area by instrumented car, Noro et al. (2015) observed consistent temperature differences in the range 0.5°C to 2.5°C depending on the Local Climate Zone (LCZ, Stewart and Oke 2012) passed through. Shi et al. (2018) confirmed the applicability of mobile sensors in assessing the thermal properties within high-density heterogeneous urban contexts, evaluating an Intra-LCZ air temperature difference up to 2°C within 6 diverse LCZs. In agreement with traditional studies on the local-scale UHI effect (e.g., Di Sabatino et al. 2020), these studies support strategies that increase vegetation coverage at the expense of buildings to mitigate urban warming and create a comfortable thermal environment. Other uses of mobile sensing include the assessment of the impacts of the urban morphology on the cooling effect of small rivers (Park et al., 2019), and the influence of external factors on the temperature field within the urban context (Rajkovich and Larsen, 2016). The temperature maps obtained through mobile sensing are also suitable for validating numerical simulations (Hsieh et al., 2016) and for application to thermal comfort and local climate stress (Koopmans et al., 2020).

The advantage of a mobile sensor is the large spatial coverage ensured by continuous monitoring while moving, which can be performed actively through ad-hoc experiments or passively during daily life activities. As a drawback, measurements will be dependent on both time and space, revealing non-trivialness to assess phenomena such as the UHI effect. Schwarz et al. (2012) introduced a correction for decreasing temperatures due to progressing time so that they would not confound air temperature differences due to changing surroundings with temperature differences because of evening cooling. To compensate for the different time responses of the mobile sensor related to the reference, namely the thermal inertia error, Qi et al. (2022) introduced an initial temperature correction. In previous research, the response time of mobile sensors was determined by cycling through a tunnel (Brandsma and Wolters, 2012) or by sensitivity tests comparing in-situ and mobile measurements (Emery et al., 2021). To deal with the response time at the beginning of a monitoring session, we can use statistics to eliminate the time required by the sensor to adjust its internal temperature to the ambient air. Depending on the scope of the investigation, a temperature decline correction is needed to compensate for the background temperature evolution while completing the route (Brandsma and Wolters, 2012). Finally, mobile sensors need a protocol for outdoor validation before usage, which is missing in most applications from the literature. For low-cost sensors, this step is required owing to the discrepancy often observed between the ideal accuracy of the sensor (that acquired in the laboratory under controlled conditions) and the real one evaluated in the field. Establishing an outdoor protocol for low-cost sensors is mandatory to infer the reliability of their measurements and under which circumstances they perform at their best (Brattich et al., 2020). Similarly, research-grade instrumentation is mostly built for fixed-location monitoring, and a reliability test should be performed for mobile usage.

In this paper, we provide a quality assessment and performance evaluation of a recently developed commercial mobile sensor for monitoring the urban microclimate, called MeteoTracker (MT) developed by IoTopon Srl. To the best of our knowledge, MT was used in very few scientific researches with promising results (Cecilia and Peng, 2022; Carraro, 2022). However, these evaluations focused only on the air temperature data of a single MT mwith respect to other low-cost mobile sensors, leaving gaps in the overall performance evaluation. In the current paper, we aim to provide a more robust and comprehensive assess-





ment of the MT outdoor performance, which includes evaluating the intra-sensor variability of multiple MTs simultaneously
operating under equal ambient conditions and validating MT's measurements against a research-grade reference in both opti-
mal (while moving) and non-optimal (as a fixed station) operational usage. This investigation also provides a set of exploitable
methods to correct MT measurements and enhance their outdoor accuracy, validated for different climate zones and seasons,
sensor usage, and customized for each measured variable of the MT. Ultimately, this analysis will evaluate the potential of the
MT to be adopted as a research-grade sensor.

After this introduction, Sect. 2 introduces the MT sensor and data flow; Sect. 3 describes the multi-step procedure adopted
in this paper to address the sensor performance while Sect. 4 presents the results from each step. Section 5 contextualizes the
MT performance in the context of mobile sensing. Section 6 draws the conclusions.

## 2  The MeteoTracker

### 2.1  The sensor

The MT is a low-cost portable weather station (see Fig. 1), that samples several meteorological variables while moving jointly
with its carrying vehicle (mobile sensor). The device hardware comprises a compact case (75 mm × 75 mm × 35 mm)
with a magnetic base to secure the station on the vehicle, tested to regulatory speed limits on highways. The case can also
support the installation of a string to secure the station to a non-metal moving object. The aerodynamic shape supports the
stability above the vehicle, while the extensive frontal and back overtures (air filters) enable a large air volume sampling and
good internal ventilation. The sensing board is supplied by different capacitive-resistive sensors, measuring air temperature
$T$ and relative humidity $RH$, and atmospheric pressure $P$ with a declared accuracy close to a research-grade instrument (see
Table 1). In addition, the sensor measures the radiation intensity $R$ but accuracy and operational range are not provided by
the manufacturer. Derived quantities are also automatically computed by the station, using known empirical thermodynamic
laws and formulas. Among those, the dew-point temperature $T_d$ and the humidex index HDX (an index which estimates the
anthropogenic well-being associated with climate, see Masterton and Richardson 1979) are both obtained by combining air
temperature and relative humidity in some capacities. For the dew point, Lawrence (2005) explored the intricacies and possible
empirical, theoretical, and simplified expressions with the classical thermodynamics, while humidex is defined as

$$\text{HDX} = T + 0.5555(e - 10) \tag{1}$$

where $e$ is the water vapor pressure in hPa and $T$ is in degrees Celsius. Altitude $Z$ in meters above the mean sea level is also
derived from the atmospheric pressure; with it, the vertical thermal gradient is computed in $^\circ\text{C}\,(100\,\text{m})^{-1}$.

The sensor is not shielded from solar radiation. Still, it is supplied with a Radiation Error Correction System (RECS) a
patent of the manufacturer to correct the effect of solar radiation on temperature while the sensor is moving at more than 7
km h$^{-1}$. The station is remotely controlled using a customized application on the user's smartphone, through a one-to-one
connection (one station is controlled by one application). Through the app, the user selects the sampling rate of the sensors
both in frequency (at least 1 Hz) and distance (at least 1 m), letting the device select and use the most crammed depending on





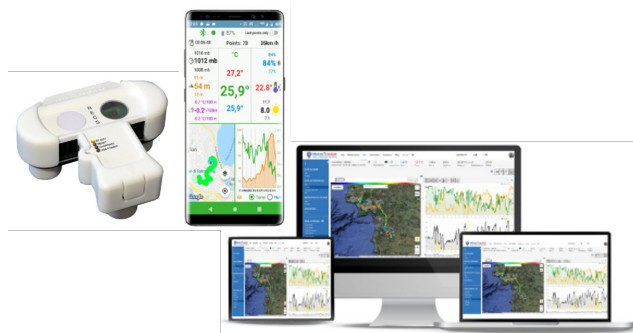

**Figure 1.** The MT and its components: left to right, the mini-weather station, the mobile application, and the web platform. Collaged from https://meteotracker.com/ (accessed on July 2022).

**Table 1.** Significant variables measured by the MT, with accuracy and operational range according to the manufacturer.

| Variable | Accuracy | Operational range |
| --- | --- | --- |
| Air temperature $T$ | $\pm 0.5°$C under solar radiation and $V_S > 7 km h^{-1}$ | $-40°$C $- +125°$C |
| Relative humidity $RH$ | $\pm 2\%$ | $0\% - 100\%$ |
| Atmospheric pressure $P$ | $\pm 3\ Pa$ (relative) or $\pm 50\ Pa$ (absolute) | $-$ |
| Altitude above mean sea level $Z$ | $\pm 10\ m$ (for the initial altitude value only) | $-$ |

[1]RECS, patent of the manufacturer

the vehicle's speed. The user decides to enable or disable the sensor calibration at each start and stop of the vehicle, and the temperature correction due to the sensor movement (to discern the sensor usage between mobile and in a fixed location). The app uses the smartphone GPS to geolocate the station and compute the initial altitude (see Sect. 3) and provides the vehicle speed $V_s$. On the app, the user can visualize the live streaming of the monitoring session, from a georeferenced map and the time series of the measured variables. At the end of each session, measurements are stored locally in the smartphone and uploaded on a dedicated online platform for visualization, data retrieval, download, and sharing with other users.

## 2.2 Data flow and Visualization

The mobile app and smartphone connectivity regulate the flow of the data collected by the station. The station and mobile app have to remain connected via Bluetooth for the whole duration of the monitoring session, to ensure the flow and storage of the measurement (the station itself does not have an internal memory). A stable GPS and internet connection are required to compute the altitude during the monitoring session, to live stream the monitoring session through the dedicated online dashboard, and to upload the data on the platform once over. The user can define the level of privacy (private, public, and public anonymously) of the collected data before initiating a monitoring session. Both public sessions will be streamed and data shared through the platform. Public data on the platform can be reached by other users through a dashboard, with permission



to visualize or download depending on the type of account bought alongside the device. An example of data visualization on
the dashboard is shown in Fig. 2. The MT track is superimposed on a Google map, showing the point-by-point georeferenced

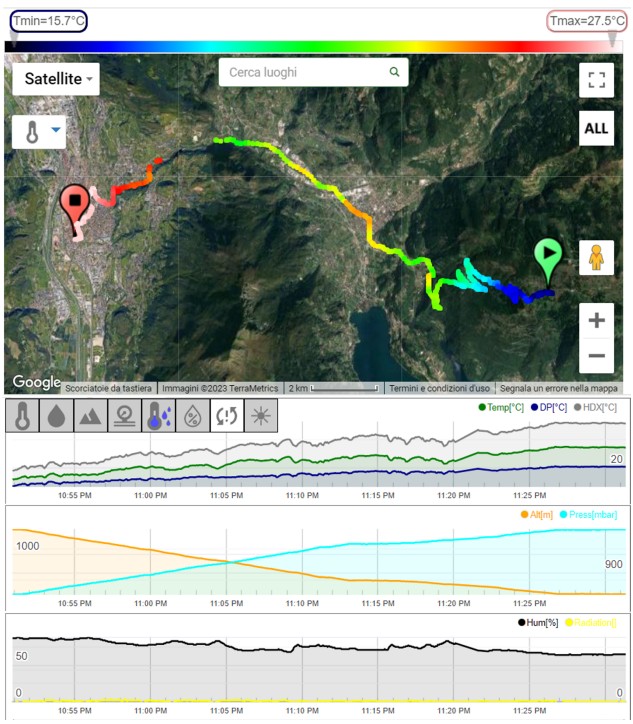

**Figure 2.** Example of MT track and collected data as visualized on the online dashboard. The coloured points on the map are the air temperature measured by the sensor. Graphs are the reconstructed time series of each measured and derived variable along the vehicle track. Source: https://app.meteotracker.com/ (accessed on July 2022).

map of a measured variable (air temperature in the example of Fig. 2), giving a qualitative but effective intake of the collected measurements. The time series of the same variables can be displayed directly on the dashboard for a more quantitative disposal of the data.

The price of the device with full permission on the platform in 2021 was 119 euros (including taxes); the recent updates (2023), the new connectivity tools, and the set of additional accessories have raised the costs of this sensor, which remains below 400 euros.

## 3   Reliability Assessment: a Multi-step Validation Process

The validation process of MT measurements is performed in four steps: (i) identification and removal of the adjusting pe-
riod to the environmental conditions, (ii) evaluation of the intra-sensor variability under similar operational conditions, (iii) confrontation with a reference station in fixed-location mode, (iv) confrontation with a reference station on the move. Three





different cities hosted one or more steps of this assessment, namely Bologna (steps i and ii) and Genova (step iii) in Italy, and Wageningen (iv) in the Netherlands, testing the sensors in different climatic areas. Each of these steps is a key element to assess the self-consistency of the instrumentation and evaluate its use for research purposes. The process evaluates the strengths and weaknesses of the MT, allowing us to expose the reliability limit of this specific low-cost sensor but also the study and computation of possible post-processing solutions to improve the data quality and usability.

### 3.1 Outlier Removal and Identification of the Adjusting Period

As a fundamental step in data post-processing, the literature has proposed several methods to identify and remove outliers from a dataset. Typical methods are based on the a-priori assumption that the dataset can be divided into subperiods wherein the measurements are distributed following a known function (typically a Gaussian distribution). A well-known example of these methods is given by Vickers and Mahrt (1997), where the outlier is defined as a value larger than 3.5 standard deviations from the mean in a certain time interval. Using this method over 30-minute windows, Barbano et al. (2021) obtained a reliable cleaning performance for fast-sampling meteorological data within the urban canopy. Working with low-cost sensors for air quality and thermal comfort detection, Brattich et al. (2020) applied the Hampel filter (Hampel, 1974) to 1-min data to detect value beyond 2 median absolute deviations over 7-min windows and replace them with the median over the same interval.

The aforementioned methods are tuned on traditional measurement techniques, where instruments sample meteorological or air quality data for a "long" period and at a fixed location. We can infer that monitoring sessions with MTs will be very short in time (as long as the vehicle travels) and frequent, not providing a sufficient amount of data to make a-priori assumptions on their distribution. Moreover, we can assume that after each monitoring session, the user would unmount the MT from the vehicle and bring it into an indoor environment for security reasons. At the onset of the next session, the MT will likely need a certain time to adjust to a change of location, i.e., a change in the ambient temperature and relative humidity. The Inter-Quartile Range (IQR) outliers' removal method (Hubert and Van der Veeken, 2008) serves both purposes of removing the outliers and the initial adjustment period. This method applies to the entire sample and it is not based on any assumption about the data distribution. It is based on the definition of upper and lower limits beyond which values are classified as outliers. Specifically, any data point $x$ is an outlier if

$$x < Q_1 - 1.5 IQR, \tag{2}$$
$$x > Q_3 + 1.5 IQR, \tag{3}$$

where $Q_1$ and $Q_3$ are the first and third quartiles, respectively, and $IQR = Q_3 - Q_1$. When needed, the IQR method is pre-empted by a linear detrending. The outliers identified at the beginning of the session mark the adjusting period and are just removed. Outliers given by spikes within the session are instead replaced with the median of the data distribution.

### 3.2 Intra-Sensor Variability

Low-cost sensors have proven reliable and accurate under laboratory conditions, yet show larger inconsistencies when used outdoors under real-world atmospheric conditions. This is due to the fast transitions and heterogeneity of the atmospheric



conditions compared to the rather constant and homogeneous laboratory flow. The response of low-cost sensors to the natural oscillations of the atmospheric variables can cause discrepancies between sensors' measurements, due to the different sensor responses. The intra-sensor variability test is an open-air experiment, where multiple sensors operate under the same environmental condition to infer the consistency among different sensors.

To the scope, three monitoring sessions were designed to operate multiple MT simultaneously (in groups of 6, 6, and 8 MTs for practical reasons). During each session, the MTs within each group were mounted on top of a designated electric car and controlled by an equal number of smartphones by the passenger. The MTs were placed side-by-side in the front part of the car's top, as close as possible to the car axis to avoid capturing flows from lateral edges. MTs aligned perpendicularly to the car axis along a single line: this prevents mutual shielding and exposes each MT directly to the flow. The spacing between MTs was approximately twice the lateral dimension of a single MT. A 50–60-min drive around the city center and outskirts is then performed during each session, starting and ending at the Department of Physics and Astronomy of the University of Bologna (44°29′57.1″N 11°21′13.8″E) and passing through different neighborhoods and local ambient conditions to capture most of the morphology variability of the city. Table 2 summarizes the general information on the sessions. The three sessions were

**Table 2.** Overview of the intra-sensor variability tests

| Session ID | Initial date and local time | Duration | Sensors |
|---|---|---|---|
| S1 | 30/06/2022, 14:30 | 60 min | 8 |
| S2 | 30/06/2022, 16:00 | 50 min | 6 |
| S3 | 01/07/2022, 12:45 | 50 min | 6 |

analyzed independently due to the change in the environmental conditions, start-and-stops that occurred during the drive, and slightly different trajectories adopted. Thus we adopted a first session to tailor our postprocessing schema on the measurements while testing those on the remaining sessions.

## 3.3 Fixed-location Comparison

The reliability and trustworthiness of low-cost sensors are typically at stake when used outdoors as large and sharp transitions in the environmental conditions (sudden temperature drops, wind gusts, etc.), as well as extreme regimes (saturation, rain, and snowfall, etc.), are often demanding for sensor accuracy. For these reasons, a comparison with research-grade instrumentation is convenient to test the actual accuracy of low-cost sensors outdoors. Differently from most of the instrumentation designed for research purposes, the MT operates on the move, introducing a further degree of complexity to the validation procedure. To counteract this limitation, a two-step validation is provided, first using the MT as a fixed-location weather station (Sect. 4.3) and then comparing its performance against a previously validated mobile station (Sect. 3.4). For the fixed-location comparison, an MT was placed behind a Stevenson screen located on the rooftop of CIMA Research Foundation headquarters (44°17′59.2″N 8°27′06.6″E) close to a reference weather station of the Acronet network. Including the sole sensors required for this study, the reference station is equipped with a shielded transducer (t026 TTEPRH, SIAP+MICROS S.p.a., Italy) sampling air temperature





in the range -30°C to +60°C with an accuracy of ±0.1°C, and relative humidity between 0% and 100% with an accuracy of 2%. The Stevenson screen is required to shield the MT from solar radiation and minimize the sensor overheating, thus replicating the work of the RECS when the sensor is moving, to expose the MT to the same environmental condition of the reference weather station, as deployed in Fig. 3. The data acquisition for this analysis covered two separate periods of continuous measurements

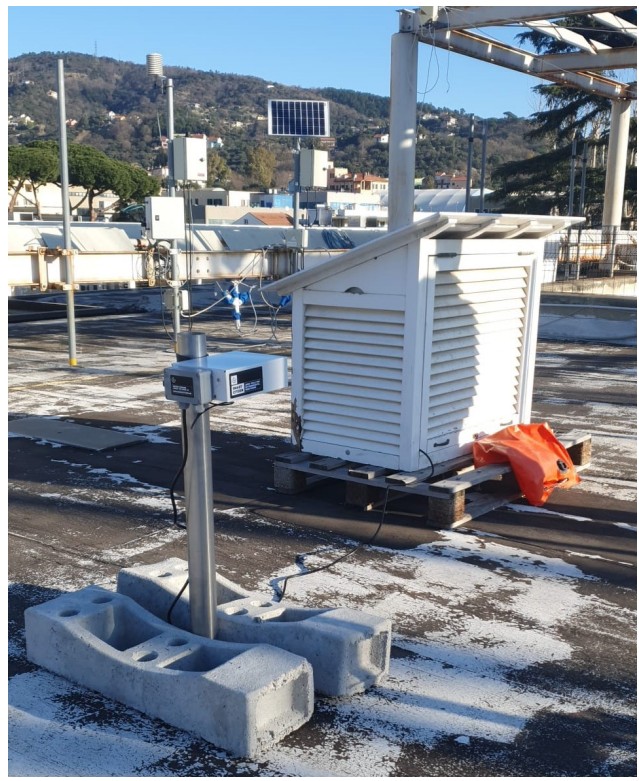

**Figure 3.** Stevenson screen (front right) and reference weather station (middle back) locations on the building rooftop.

lasting approximately one month each: from 19 December 2022 to 10 January 2023 (winter period), and from 11 July to 3 August 2023 (summer period). No weather conditions were discharged from this analysis.

### 3.4 Mobile Comparison

To test the MT under conventional operation conditions, two sensors were mounted on a meteorological cargo bike (see Fig. 4a) developed at Wageningen University (The Netherlands) to measure air temperature, relative humidity, wind speed, and
radiation in a reliable way (Heusinkveld et al., 2014). The temperature and relative humidity are measured with a shielded thermometer-hygrometer (model CS215L, Campbell Scientific, USA) with a radiation screen mounted at 1.2 m. Heusinkveld et al. (2014) ensures an accuracy of less than 0.1°C for air temperature and 2% for relative humidity (within the range 10-90%). Wind speed and direction are measured with a Gill WindSonic (Gill Instruments, UK), and used to derive the ventilation speed,





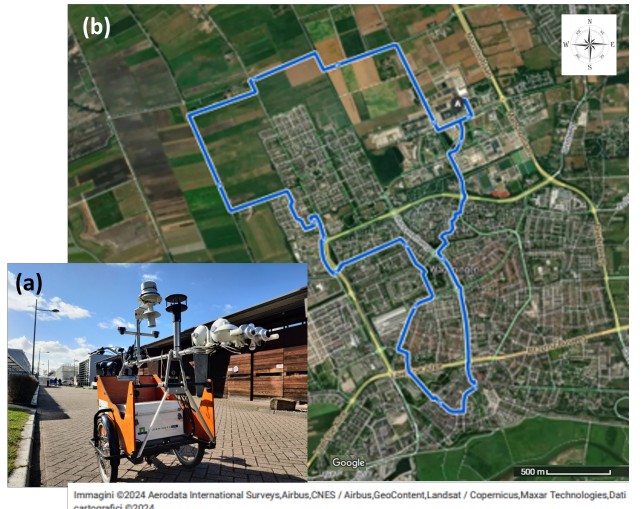

**Figure 4.** A picture of the cargo bike with its equipment (a) and the cycling route operated for comparison (b)

i.e. the speed of the air sampled by the sensor and given by the vectorial combination of the wind and cargo-bike speeds.

The MTs were placed unshielded at the same height as the reference temperature sensor, one to its left and the other to its right. With this setup, the average temperature of the MTs corresponds to the reference temperature. The reference temperature sensor samples at 1 Hz; with an average cycling speed of 4 $\mathrm{ms^{-1}}$ it corresponds to a sample every 4 m. The MTs measure with a 3-s sampling rate, resulting in a sample approximately every 12 m. A block average of 5 s is chosen for the comparison.

The data collection was performed in 8 sessions from January $11^{th}$ to March $3^{rd}$, 2023, lasting around 1-h each between

12:00 and 18:00 local time (except for a single evening session starting at 20:30 local time). The dataset encloses different atmospheric conditions, mostly accounting for sunny days when incoming radiation varies the most, or partially cloudy conditions (with no precipitation). The cycling track followed a fixed route, with a seemingly constant cycling speed of 4 $\mathrm{ms^{-1}}$. This cycling route went through an open grassland area near Wageningen (Binnenveld), a new residential area (Wageningen Noordwest), and the historic city center of Wageningen (see Fig. 4b).

## 4 Results

### 4.1 Removal of Outliers and Adjusting Period

From a visual inspection of the data, there are no clear outliers inside the measuring periods. The stability of the MT and its sampling rate minimize the possibility of spikes in the data acquisition, actually preventing the occurrence of outliers within the time series. Conversely, there is clear evidence that upon activation, the sensor undergoes a process of adjustment to align with

the surrounding ambient temperature. Several preliminary tests were implemented during wintertime to investigate the MT's adjusting times according to the different ambient conditions the MT has to adjust to, and the cooling rate it has to endure. All





these tests were performed by bicycle in Bologna. Equation 3 is applied to isolate and remove these adjusting periods from the data distribution of this test. Figure 5 picks an example session to show the application of the outlier removal method. The

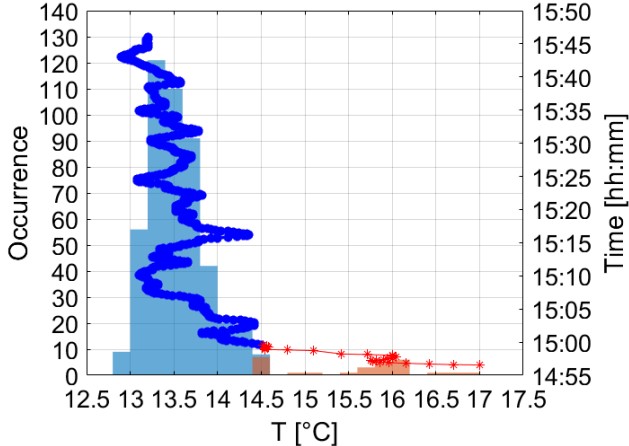

**Figure 5.** Distribution by the occurrence of the air temperature for the example session, alongside the time evolution of the session. The colors highlight the removed adjustment time after the application of Eq. 3 (blue) compared to the filtered data distribution and time series (red)

distribution of the air temperature follows a skewed normal distribution, with a bimodal factor when outliers are accounted

for. The outliers compose the adjustment period and no further spikes are detected through the time series. The maximum temperature recorded at the beginning of the session in Fig. 5 is 17.0°C, falling to 14.5°C after applying Eq. 3. Similarly, the initial minimum humidity level is recorded at 52%, but it increases to 59% during the same time frame of the temperature drop after applying Eq. 3 (not shown).

Table 3 summarizes the average temperature changes, adjusting times, and cooling rates evaluated through the preliminary

tests and the mobile comparison test performed with the cargo bike (Sect. 3.4). Temperature changes and adjusting times are direct outcomes of the outlier removal, while the cooling/heating rates are computed as their ratios. The discrepancy between adjusting periods and cooling/heating rates is owed to the initial air temperature and the different times of the day, and so the intensity of solar radiation. Natural ventilation can also play a role in increasing the cooling rate. Overall, the cooling/heating rate of an MT under operational use ranges between 0.5°C min$^{-1}$ to 1.7°C min$^{-1}$, accounting for an adjustment time of

2.2–6.3 min according to this test. Table 3 only reports the application of the removal method on the air temperature but the same procedure has been independently applied to the relative humidity, obtaining similar results. For this reason, temperature alone is taken as a benchmark for identifying the adjustment period.

To provide further support for the acclimatization process of the MT, supplementary comparison tests with an analog thermometer are carried out both indoors and outdoors. Results of these tests suggest that a stationary MT takes 15 minutes (on

average) to achieve thermal equilibrium in a partially controlled room from an initial temperature discrepancy of 2°C. When exposed to outdoor conditions, the stationary MT takes 10 minutes (on average) to drop its temperature of 2°C. Thanks to the



**Table 3.** Average air temperature, temperature drop (in absolute value), adjusting time, and cooling rate for the different mobile monitoring sessions involved in the tests. For each session, the date, starting time, and duration are also reported. Comp sessions are performed in Bologna, Cargo ones are from the mobile comparison in Wageningen. Multiple dates and start times refer to the single days and beginning of each session whose values are averaged

| Session | Date and Local start time | Duration [min] | Initial $T$ [°C] | $T$ drop [°C] | Adjustment Time [min] | Cooling rate [°C min$^{-1}$] |
|---------|---------------------------|----------------|------------------|---------------|------------------------|------------------------------|
| CompA | 03/03, 13:59 | 38 | 16.6 | 2.6 | 5.9 | 0.5 |
| CompB | 03/03, 14:56 | 49 | 17.0 | 1.7 | 2.7 | 0.6 |
| CompC | 03/03, 15:56 | 47 | 18.8 | 4.8 | 2.8 | 1.7 |
| CompD | 20/03, 14:12 | 28 | 16.7 | 2.6 | 3.2 | 0.8 |
| CompE | 20/03, 17:24 | 46 | 14.9 | 1.7 | 2.2 | 0.8 |
| CargoA | 11/01, 12:00-16:00 | 55 | 11.7 | 1.4 | 2.2 | 0.6 |
| CargoB | 18/01–07/02, 12:00-14:00 | 55 | 9.4 | 5.2 | 5.5 | 0.9 |
| CargoC | 07/02–03/03, 16:30-20:30 | 55 | 13.3 | 6.3 | 5.2 | 1.2 |

thermal regulation induced by ventilation, the adjustment times reduce largely when the sensor is moving at a speed higher than 7 km h$^{-1}$. For the intra-sensor variability test and the fixed station comparison, we waited 15–20 minutes after having prepared the experimental setup, letting the MT adjust to the ambient air.

## 4.2 Intra-Sensor Variability and Data-Correction Methods

The results of the intra-sensor variability analysis are presented alongside the development and application of data-driven correction methods for data postprocessing. The scope of the corrections is to decrease the variability of data around the mean and increase the linear relationship of each track to the mean within the session, thus reducing the intra-sensor variability. In particular, a correction was searched and applied wherever single tracks were consistently over- or under-shooting the range

represented by the instrumental error applied to the session average. Session S1 is used as a test case, while S2 and S3 are the benchmark. Data measured by the MTs are averaged every 5 s to remove the discrepancy in the time-space acquisition dichotomy and to homogenize the dataset. For each variable, the average session is computed to explore the stability of each sensor measure around the mean. The comparison for the measured variables is shown in Fig. 6. As a byproduct, the comparison masks the non-systematic errors while enhancing the existing biases between different measures. This results in large values of

the determination coefficient despite the single sensor measurements occasionally being well beyond the range defined by the instrumental error around the average session. This is the case of relative humidity (Fig. 6a) and less so of pressure (Fig. 6c). While for the pressure the discrepancy between different sensors accounts for up to 100% of the instrumental error, that for relative humidity is up to 500%. Pressure displays a non-homogeneous distribution of values around the average session, but rather a step-wise measurement trend, probably owing to the low sensitivity of the sensor. Air temperatures fall mostly within

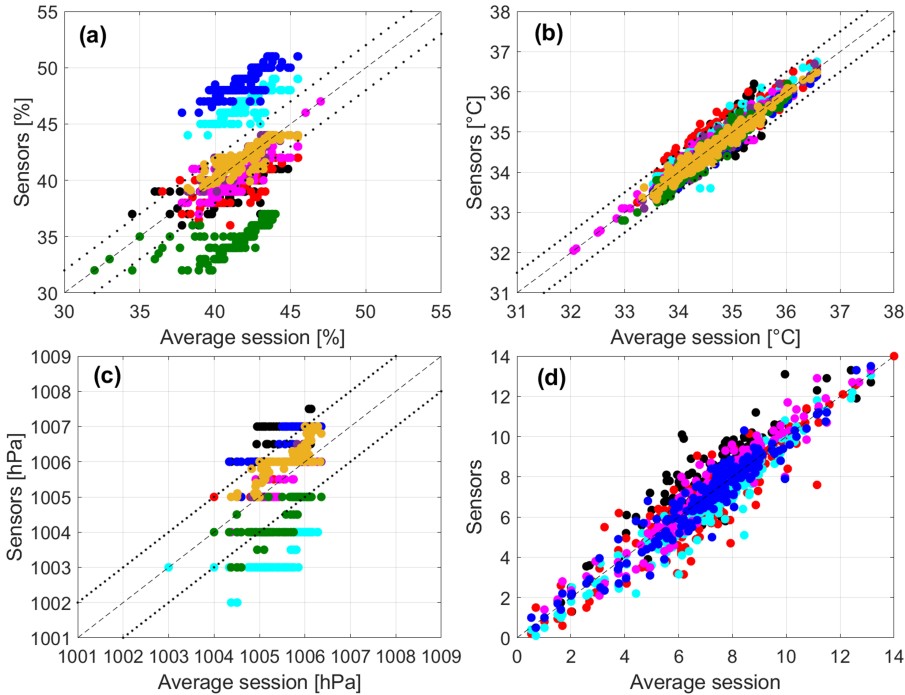

**Figure 6.** 5 s averages of relative humidity $RH$ (a), air temperature $T$ (b), barometric pressure $P$ (c), and radiation intensity $R$ (d) measured by each sensor as a function of the average session. Each color identifies the measurement of a single MT. The dashed line is the bisector, the dotted lines are the instrumental error around the bisector

the instrumental error (Fig. 6b), apart from a few values in two sessions. Air temperature is considered to perform at a level of accuracy in line with the manufacturer's indication and will not undergo any corrections. The radiation intensity (Fig. 6d) provides a measure of the intensity of solar radiation, with sensors being affected by different shadowing along the track. The performance of this quantity is qualitatively appreciable despite the lack of information on the instrumental error preventing a more quantitative evaluation.

From the measurements described above and the information retrieved from the GPS of the smartphone, the sensor automatically derives several quantities including altitude $Z$, vehicle speed $V_s$, the dew point $T_d$ and the humidex index HDX (see Fig. 7). The vehicle speed is self-consistent, despite an increasing spread of values at low velocities (Fig. 6d). Despite an instrumental error for the vehicle speed is not provided by the manufacturer (as it depends on the smartphone the sensor is connected to) and a more quantitative assessment of this variable is precluded, it can still be used as an indicator of the stability for different smartphone models. As for the altitude, the phone GPS is involved in the computation of the vehicle velocity, thus similarities between measurements are expected but not granted.

Altitude above the mean sea level (Fig. 7c) is automatically computed by the MT by retrieving the initial altitude $Z_0$ from an open web server using the location of the initial latitude and longitude and then modifying its value along the track according



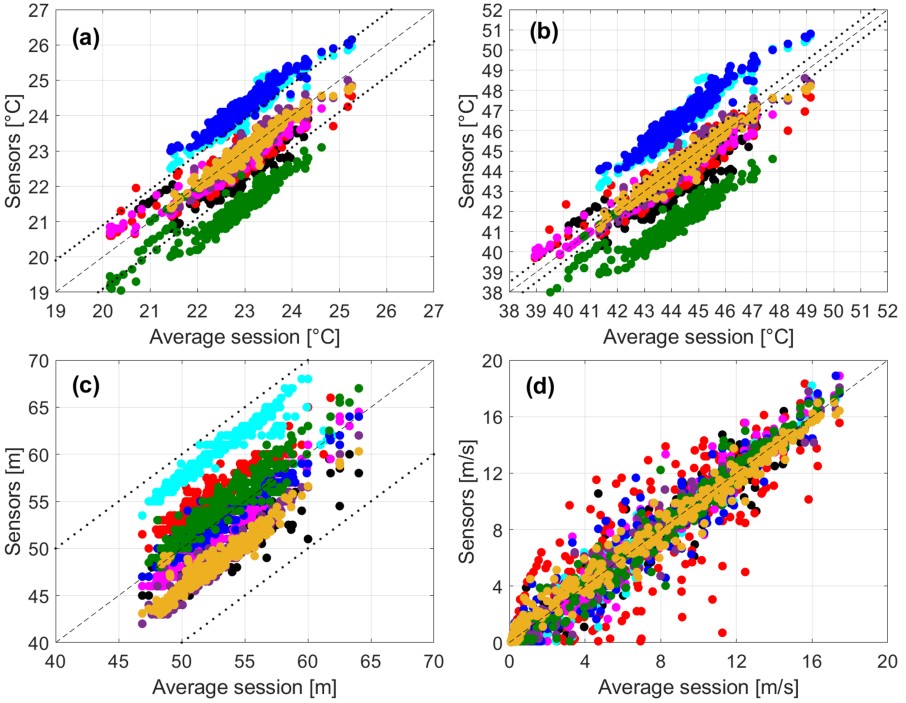

**Figure 7.** 5-s averages of the dew point $T_d$ (a), humidex HDX (b), altitude above mean sea level $Z$ (c), and vehicle speed $V_s$ (d) retrieved from each sensor as a function of the average session. Each color identifies the measurement of a single MT. The dashed line is the bisector, the dotted lines are the instrumental error around the bisector

to the measured pressure using

$$Z = \frac{T_0}{\Gamma_s}\left(\left(\frac{P}{P_0}\right)^{-\Gamma_s R_d/g} - 1\right) + Z_0 \tag{4}$$

where $T_0$, $P_0$, and $Z_0$ are the air temperature, pressure, and altitude collected as the first data measure, respectively, $\Gamma_s = 0.0065°C\ m^{-1}$ is the standard atmospheric lapse rate, $g = 9.81\ m\ s^{-1}$ is the acceleration due to gravity, and $R_d$ is the gas constant for the dry air. The nominal error of $\pm 10$ m associated with the altitude measurement is therefore a composition of the error propagation given by Eq. 4 and the GPS accuracy. It is worth noting that the accuracy of the GPS is a property of the smartphone connected to the MT, and it can change according to the smartphone brand and model. Since this test was conducted using different smartphones we can have introduced an additional source of error. The correction we are about to introduce will minimize this error but it is arguable that using the same smartphone model and brand the correction would not be needed (at least if the error from Eq. 4 is negligible). The large nominal error allows a good altitude performance within the current test, but biases between sensors are also evident. For this reason, an alternative procedure to compute altitude is proposed. It consists of the retrieval of the altitude from a preferred web server (as the one used in this paper) or a digital elevation model map for each latitude-longitude couple measured by the GPS, thus bypassing Eq. 4. In other words, the sole





error remaining in the computation of the altitude is associated with the GPS signal. This procedure is more computationally expansive than Eq. 4, but it ensures a better evaluation of the altitude as long as the GPS signal is stable. Applied to S1, the method grants a large improvement in the altitude computation, with a drastic reduction of the sessions' spread around the mean one (Fig. 8). The improvement in the altitude values enables the correction of pressure. From Eq. 4, we can derive an

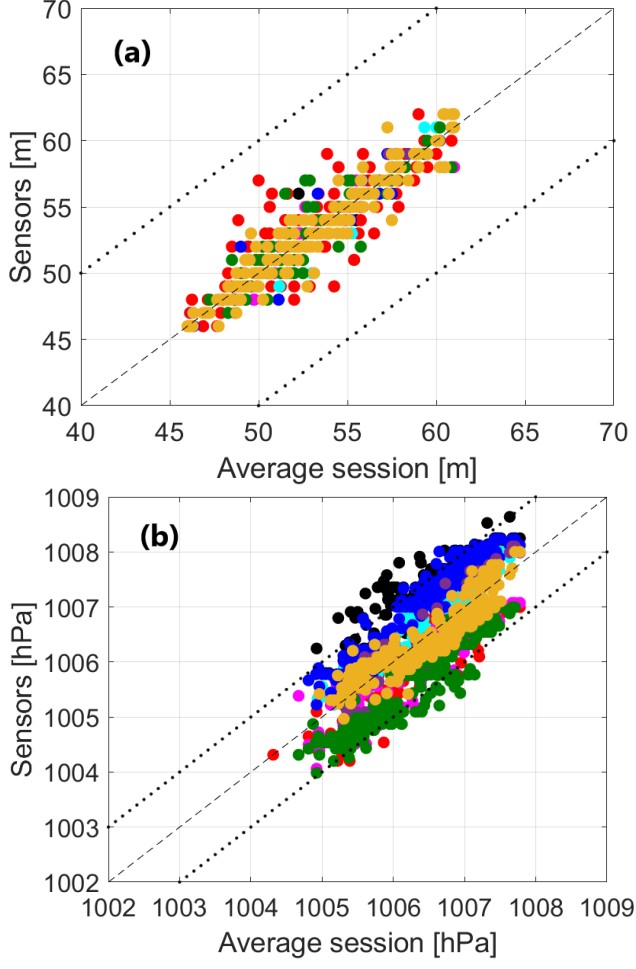

**Figure 8.** 5 s averages of the corrected altitude (a) and pressure (b) as a function of the average session. Each color identifies the measurement of a single MT. The dashed line is the bisector, the dotted lines are the instrumental error around the bisector

310

expression for pressure $P_c$ as

$$P_c = P_0 \left( \frac{\Gamma_s}{T_0} \left( Z_c - Z_0 \right) + 1 \right)^{-g/(\Gamma_s R_d)} \tag{5}$$

where $Z_c$ is the altitude above the mean sea level as retrieved from the web server. The recomputed pressure is shown in Fig. 8. The correction collapses pressure from different sensors within the instrumental range of uncertainty (around the average





session) while introducing a richer value distribution. This is supposedly caused by the larger sensitivity of the GPS sensor compared to the barometer.

Using $P_c$, we can estimate a first correcting formula for the relative humidity. The relative humidity is defined as the ratio between the water-vapor pressure $e$ and the saturation water-vapor pressure $e_s$. Assuming that the variability of the total pressure is entirely driven by that of the water vapor, $\delta RH \propto \delta e$. The application of this correction type to the relative humidity revealed almost negligible for the dataset under investigation and for this reason, it was discarded. Nonetheless, $e$ and $e_s$ are known to be functions of the dew point $T_d$ and the air temperature $T$ respectively, owing to a large number of empirical (e.g., the Magnus formula) and theoretical (derived from the Clausius-Clapeyron equation) expressions. The link between $T$, $T_d$, and $RH$ suggests finding possible correction methods for $RH$ based on these variables, to which the humidex index HDX can be added owing to its dependency on both temperature and relative humidity. Indeed both $T_d$ and HDX show an odd intra-sensor variability which is propagated from the relative humidity (see Fig. 7a,b), with the dew point being affected the most. As from its definitions in Eq. 1, the humidex index depends more on the air temperature and less on relative humidity (or dew point). In the range of temperature where the humidex is defined, the adding term on the RHS of Eq. 1 is much smaller than $T$, and thus the intra-sensor variability in the humidex is less pronounced than $T_d$ and $RH$. The humidex index can be used as a starting point for the correction procedure which ultimately will allow to correct $RH$, $T_d$ and HDX. Practical definitions of this index can be retrieved from Eq. 1, following the standard used by Environment and Climate Change Canada

$$\text{HDX} = T + 0.5555 \left( 6.11\text{e}^{\left( L/R_w \left( \left( \frac{1}{273.16} \right) \left( \frac{1}{273.15+T_d} \right) \right) \right)} - 10 \right) \tag{6}$$

where $L$, the latent heat of vaporization, is retrieved from the linear interpolation of $T$ knowing that $L(T = 0°\text{C}) = 2.501 \cdot 10^6$ J kg$^{-1}$ and $L(T = 100°\text{C}) = 2.257 \cdot 10^6$ J kg$^{-1}$, while $R_w = 461$ J K$^{-1}$kg$^{-1}$ is the gas constant for a moist atmosphere. Alternatively, we can adopt some Magnus formula for $e$, so that

$$\text{HDX} = T + 0.5555 \left( 0.06RH10^{0.03T} - 10 \right). \tag{7}$$

Both empirical equations require $T$ and $T_d$ in °C. Figure 9 shows the correlation and dispersion of the humidex calculated with Eqs. 6 and 7 and the one directly computed by the sensor. Each distribution resembles a Gaussian shape, with a high degree of symmetry and mean (and median) value close to $44°C$. Despite Eq. 7 being a better approximation to the HDX retrieved by the sensors, Eq. 6 already reduces the standard deviation of the distribution and thus the intra-sensor variability.

Substituting Eq. 6 into 7, and solving for $RH$, we obtain an expression for the corrected relative humidity $RH_c$ which only depends on the dew point (and air temperature)

$$RH_c = 100e^{\left( \frac{L}{R_w} \left( \frac{1}{273.16} - \frac{1}{273.15+T_d} \right) \right)} \left( 10^{0.03T} \right)^{-1} = 100\frac{e}{e_s} \tag{8}$$

where $e$ is formulated following Environment and Climate Change Canada and $e_s$ follows a simplified Magnus formula. From Eq. 6, we can assume that the significant component of the intra-sensor variation in HDX is given by $T_d$. That is $\delta\text{HDX} \propto \delta T_d$. Using the error propagation theory, the variability $\delta\text{HDX} = |\frac{d\text{HDX}}{dT_d}|\delta T_d$. Solving for $\delta T_d$ we get

$$\delta T_d = -\frac{R_w}{L} \left( 273.15 + T_d \right)^2 \left( 3.39 \right)^{-1} e^{-\frac{L}{R_w} \left( \frac{1}{273.16} - \frac{1}{273.15+T_d} \right)} \delta\text{HDX} \tag{9}$$





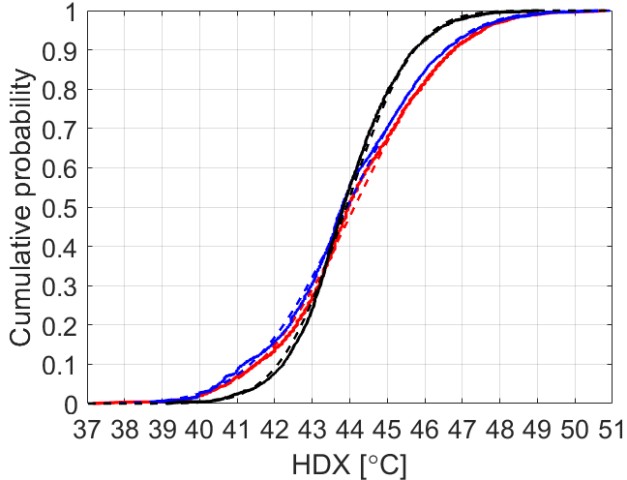

**Figure 9.** Cumulative density distribution of the HDX index computed using Eq. 6 (solid black), Eq. 7 (solid blue) and directly by the sensor (solid red). The dashed lines, paired with similar colors, represent the best normal distribution fit for each data source. The number of bins for the discretization is equal to the square root of the total number of data (equal for each distribution)

The dew point is then corrected as

$$T_d^c = T_d - \delta T_d, \tag{10}$$

and used to recompute the humidex using Eq.6. Equations 9, 10 and 6 are computed recursively to minimize the intra-sensor
differences on both $T_d$ and HDX. The minus sign in Eq. 9 is introduced to evaluate both positive and negative components of the derivative in the error propagation. After 2 iterations, the variability around the average session is already greatly reduced (see Figs. 10a,b). Note that the recursive method is intrinsically going to diverge after a large number of iterations, thus imposing a truncation after a few. Truncation is here done after visualizing the minimum variability, constrained to a range of values in agreement with the measurements. Using the recursive method, we achieved a better agreement between sensors ensuring a
smaller spread of values around the average session and the constrain of them within the instrumental errors. The instrumental error for the dew point is here derived from Lawrence (2005) approximated relation

$$T_d = T - \frac{RH - 100}{5}. \tag{11}$$

By comparing the dew point retrieved by the sensor and that computed using Eq. 11, we observe two similar distributions (see Fig. 11) suggesting we can adopt Eq. 11 for the computation of the dew point error. The agreement shown in Fig 11 raises
doubts about how solid the dew point data are when directly retrieved by the MT. Equation 11 was suggested by Lawrence (2005) to work in a moist environment, with $RH \leq 50\%$, with a possible extension to 40% introducing a further correction which resulted meaningless for the present study. Nonetheless, values below the 40% threshold were observed, for whom we can assume the approximation works just fine enough for deriving the dew-point error. In any case, this error evaluation is



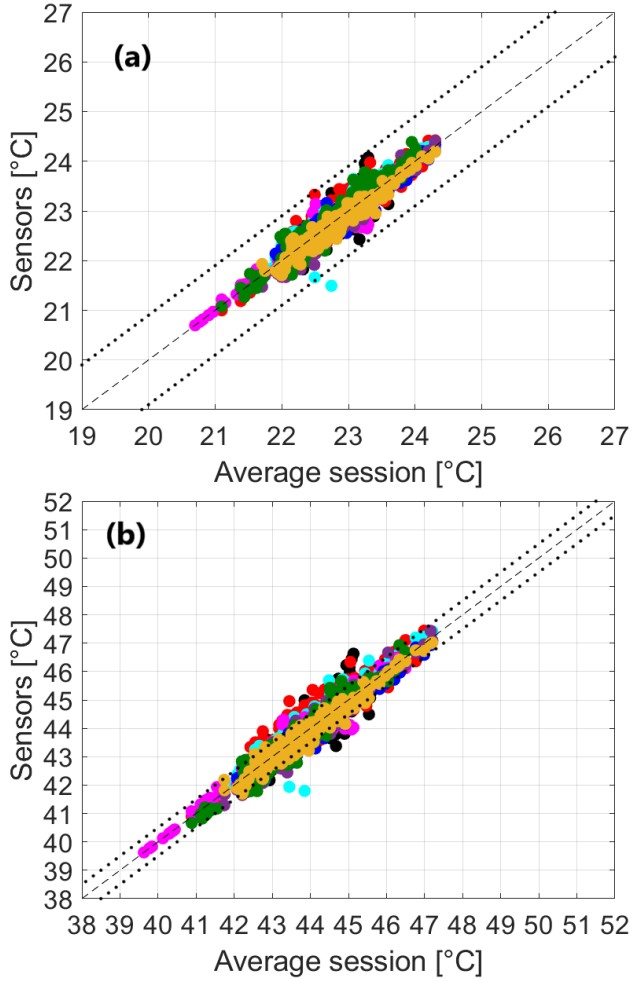

**Figure 10.** Calculated $T_d$ (a) and HDX (b) after 2 iterations using Eq. 10 and 7 as a function of the average session, respectively. Each color identifies the measurement of a single MT. The dashed line is the bisector, the dotted lines are the propagated instrumental error around the bisector evaluated using Eq. 12 for $T_d$ and 13 for HDX

limited to a medium-high humid environment and should be further tested for more arid conditions. A further assessment of

the quality of the dew point retrieved from the sensor is given in Sect. 4.3.

The error associated with the retrieved and corrected dew point reads

$$\Delta T_d = \Delta T + \frac{1}{5}\Delta RH. \tag{12}$$

In analogy, the error associated with the humidex index is computed from Eq. 6, and reads

$$\Delta \text{HDX} = \Delta T + \frac{\text{HDX} - \text{T}}{(273.15 + T_d)^2}\frac{R_w}{L}\Delta T_d. \tag{13}$$

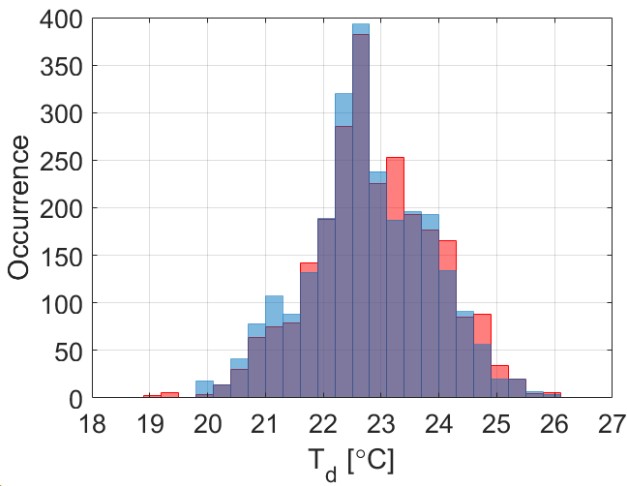

**Figure 11.** Distribution of the occurrence of the dew point $T_d$ as retrieved from the sensor (red) and computed using Eq. 11 (blue). The purple area of the graph is where the two distributions superimpose one to the other. The bin width is 0.5°C on an equal number of bins among the distributions.

As dew point and humidex converge to their minimum variability, the corrected value of $T_d$ is used to recompute the relative humidity using Eq. 8, and the resulting difference from the average session is shown in Fig. 12. Despite the correction is not

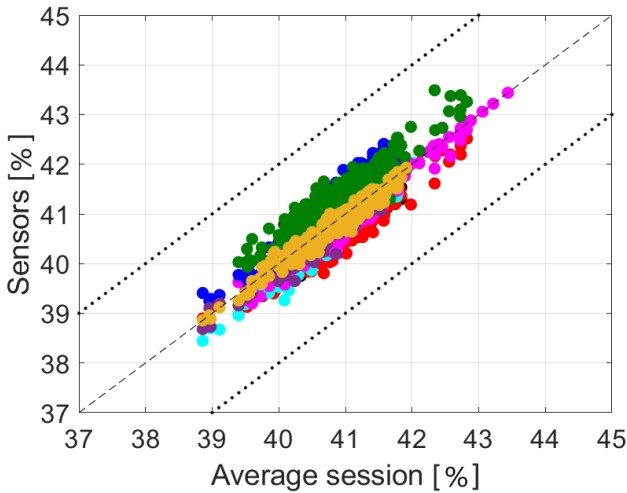

**Figure 12.** Corrected $RH$ using Eq. 8 as a function of the average session. Each color identifies the measurement of a single MT. The dashed line is the bisector, the dotted lines are the propagated instrumental error around the bisector.

always sufficient to bring the variability within the instrumental error from the average session (S2 for instance would have required an errorbar of ±3%, not shown), the intra-sensor agreement is largely improved as also quantified by the root-mean-square and mean bias errors listed in Table 4 for each session and variable. The correction removes a consistent portion of the





sensor variability, producing a maximum value of uncertainty around the average session of $\pm 1.2\%$ for S1. All the corrections

**Table 4.** Coefficient of determination $R^2$, root mean square error RMSE, and mean absolute percentage error MAPE computed session by session between the data obtained (measured, derived, or corrected) from each sensor and the respective average session. Below is reported the average $R^2$ within each session, and the maxima RMSE and MAPE

|  |  | $Z$ | $Z_c$ | $RH$ | $RH_c$ | $T_d$ | $T_d^c$ | HDX | $HDX_c$ | $P$ | $P_c$ | $T$ | $R$ |
|---|---|---|---|---|---|---|---|---|---|---|---|---|---|
| | $R^2$ | 0.85 | 0.96 | 0.61 | 0.90 | 0.85 | 0.91 | 0.87 | 0.93 | 0.58 | 0.86 | 0.94 | 0.89 |
| S1 | RMSE | 7.1 m | 1.4 m | 6.9% | 0.4% | 1.4°C | 0.2°C | 2.6°C | 0.5°C | 2.0 hPa | 0.8 hPa | 0.3 °C | 1.1 |
| | MAPE | 11.2% | 1.5% | 17.8% | 0.9% | 6.3% | 0.7% | 5.9% | 0.8% | 0.2% | <0.1% | 0.6% | 14.6% |
| | $R^2$ | 0.96 | 0.99 | 0.63 | 0.85 | 0.92 | 0.88 | 0.82 | 0.94 | 0.86 | 0.97 | 0.95 | 0.9 |
| S2 | RMSE | 4.7 m | 0.8 m | 7.3% | 1.6% | 1.5°C | 0.6°C | 3.6°C | 0.6°C | 1.2 hPa | 0.5 hPa | 0.2 °C | 0.9 |
| | MAPE | 7.9% | 0.7% | 31.9% | 3.8% | 7.2% | 2.3% | 8.5% | 1.2% | 0.1% | <0.1% | 0.4% | 9.6% |
| | $R^2$ | 0.99 | 0.99 | 0.81 | 0.95 | 0.95 | 0.97 | 0.96 | 0.98 | 0.91 | 0.97 | 0.98 | 0.97 |
| S3 | RMSE | 0.7 m | 0.7 m | 4.3% | 0.3% | 0.9°C | 0.3°C | 1.5°C | 0.6°C | 0.7 hPa | 0.4 hPa | 0.3 °C | 0.8 |
| | MAPE | 0.9% | 0.5% | 8.9% | 0.5% | 3.9% | 1.3% | 3.4% | 1.3% | <0.1% | <0.1% | 0.9% | 9.3% |

RMSE $= \left( \frac{1}{n} \sum_{i=1}^{n} \left( x_i - \langle x_i \rangle \right)^2 \right)^{1/2}$, MAPE $= \frac{100}{n} \sum_{i=1}^{n} \left| \frac{x_i - \langle x_i \rangle}{x_i} \right|$, with $x_i$ the iteration of variable $x$ from each sensor, $\langle x_i \rangle$ the average session of the same variable, $n$ the number of finite data points in $x$


we have discussed so far have substantially improved the data quality. The absolute errors also decrease with an overall increase in the determination coefficients. We have already mentioned how the intra-sensor variability here assessed, creates a bias in the coefficient of determination due to the dependence of the mean from its constituents. Nonetheless, the initial large values of this coefficient suggest that the existing discrepancy between sensors' measurements is bias-driven, while tendencies are

in accordance. The increase in the determination coefficients obtained through the correction methods explains the further reduction of the non-linear uncertainties in the sensors' measurements. The error reduction is instead an indication of the bias decrease; in other words, the sensors' measurements collapse closer to their average.

The pressure–altitude correction encloses both tendency and bias regulation effects. Pressure loses its step-wise conformation, favoring a more homogeneous data distribution which increases the linear dependence between different sensors. Altitude

decreases the intra-sensor bias by reducing the mean absolute percentage error (MAPE) of 87% and 91%, and the root mean square error (RMSE) of 80% and 82%, in S1 and S2 respectively. The iteration method involving relative humidity, dew point, and humidex is also successful in reducing both biases and tendency discrepancies, especially for the first variable. The largest effect on dew point and humidex is a decrease in the bias around the average, with an efficacy of the correction in the range 60–88% for both MAPE and RMSE, with a slightly better reduction performance in the humidex. The correction also improves

the respective coefficients of determination, but the increase is less impactful possibly due to the stronger dependence of both variables on the air temperature (which has high scores and did not need any correction) and less so on the relative humidity. Finally, the relative humidity undergoes the largest benefits from the correction method on both tendency linearity and bias,





as it is already clear from a visual comparison between Fig. 6a and 12. The linearity around the average session increases by 15-30% ensuring a more regular distribution of the data and tendency among different sensors' measure. Even more evident is the improvement by reducing the intra-sensor bias: MAPE and RMSE increase by 78–95%, which is $> 90\%$ if we exclude S2 where the correction method was not sufficient to shrink the intra-sensor variability down within the instrumental error due to the presence of an "outlier" sensor run. By removing this odd sensor run, S2 aligns with S1 and S3 statistics, allowing rigorous comparability between different MTs' measurements.

### 4.3 Fixed-location Comparison

The intra-sensor variability test has revealed the capability of the MT to perform a self-consistent assessment of the environmental conditions, even though some corrections are necessary to achieve a reasonable performance. In this and the following section, we address the performance of the MT against reference stations, being in a fixed location and under optimal usage conditions (i.e., on the move). As introduced in Sect. 3.3, sensors' comparison during both winter and summer are investigated, to verify the consistency of the previous results on both extremes of the annual thermodynamic cycle We selected periods of continuous measurements without technical issues on both apparatuses. The winter period under investigation has experienced mild weather conditions, with warm temperatures for the season and relative humidity never reaching saturation (according to the reference weather station at the CIMA Research Foundation). Moreover, neither severe nor extreme weather phenomena were observed during the period, thus facilitating the comparability between the MT and the reference weather station. Within the summer of the new global temperature record concerning the warmest July of the measurement era, Genova experienced a warm summer with daily mean and maximum temperatures above the climate average. The period included several heatwaves striking the whole region, while no severe thunder- or rain-storms were observed. The comparison is strictly limited to air temperature and relative humidity as the weather station does not measure any other variable relevant to the MT. An assessment of the dew point and humidex is also provided as both quantities are necessary for the correction of the relative humidity as we have introduced in Sect. 4.2. To strengthen the statistical robustness of air temperature and relative humidity comparisons, the coefficient of determination is computed for each variable, alongside the mean percentage error (MPE) and the RMSE to address the goodness of the linear fits. Differently from the intra-sensor variability, here the percentage error is not computed in absolute value to quantify the over- or under-estimation of the MT measurement compared to the reference one. Furthermore, we computed the MPE for the negative and positive differences separately, to better appreciate the overall over- and under-estimations by the MT. To avoid miscalculation of MPE for temperature differences across 0°C, we previously converted the air temperature in Kelvin. Willmott (1981) introduced a decomposition of the mean square error (MSE) into a systematic and an unsystematic component, separating biases from random errors. Following this philosophy, we decomposed the RMSE into its systematic

$$\text{RMSE}_\text{s} = \sqrt{\frac{1}{n}\sum_{i=1}^{n}\left(\hat{x}_i - y_i\right)^2} \qquad (14)$$





and unsystematic

$$\text{RMSE}_{\text{u}} = \sqrt{\frac{1}{n}\sum_{i=1}^{n}(x_i - \hat{x}_i)^2} \tag{15}$$

components, where $x_i$ is the iteration of variable $x$ from the MT, $y_i$ that for the reference weather station, $n$ the number of finite data points in $x$, and $\hat{x}_i = ay_i + b$ is the linear fit of $x$ on $y$. The systematic component is the consistent bias that produces the over- or under-estimation of the reference values by the MT measurement compared to the linear fit between the two datasets. The unsystematic component evaluates the scatter about the linear regression line of the MT measurements. RMSE is used instead of the MSE for a better comparison with the intra-sensor variability test and the performance evaluations found in the literature (see Sect. 5.2).

Figure 13 compares the air temperature and relative humidity of the MT and the reference station for both winter and summer periods. For both quantities, data has been previously processed and averaged every 10 minutes, and a linear fit is computed

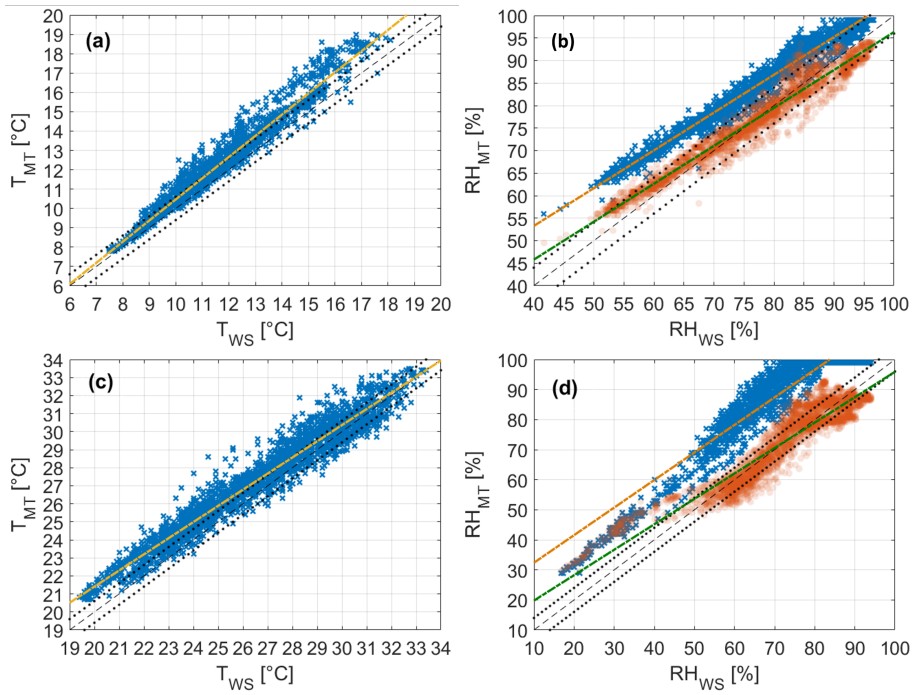

**Figure 13.** Air temperature $T$ and relative humidity $RH$ comparisons between MT (subscript MT) and reference weather station (subscript WS) measurements for the winter (panels (a) and (b)) and summer (panels (c) and (d)) periods, and relative linear fits. In (b) and (d), blue dots and orange fits refer to the measured relative humidity while red dots and green fits refer to the relative humidity corrected using Eq. 8. The dashed line is the bisector, and the dotted lines identify the range given by the sum of the MT and weather station instrumental errors around the bisector, that is $\pm 0.6°C$ for air temperature and $\pm 4\%$ for relative humidity

as a result of the comparison. During both seasons, the air temperature measured with the MT is in reasonable agreement





with the weather station, despite a clear tendency to overestimate the reference values. The wintertime overestimation also increases with increasing values of air temperature (Fig. 13a). Apart from the possible intrinsic lack of accuracy of the MT, two possible reasons may affect the performance of the sensor. First, the MT is a mobile sensor used as a fixed meteorological station, i.e. it is operated with a sub-optimal working configuration. Second, the Stevenson screen is not a perfect shield from solar radiation, whose beam can be reflected from multiple surfaces on the building rooftop into the box volume. In addition, the natural ventilation of the screen is partially prevented by its structure and air may become stagnant within its volume. It is worth noting that the flat rooftop hosting the experiment is covered with black tarpaulin, thus minimizing the reflection from one of the major sources but increasing the emission temperature of the rooftop, with a larger effect on stagnant air. As a result, the combination of inhibited ventilation and enhanced radiation might increase the air temperature within the screen volume, especially at higher temperatures when we can expect larger solar radiation. Nonetheless, the summertime overestimation is homogeneously distributed along the entire temperature range (Fig. 13c) suggesting the possible air warming inside the Stevenson screen does not increase linearly with either the air temperature or the solar radiation. Unfortunately, being the MT under the screen and the weather station are not equipped for measuring solar radiation, we can only leave these considerations as hypotheses.

The relative humidity measured by the MT largely overestimates that of the weather station (Fig. 13b,d). Given the outcomes of the intra-sensor variability test, this result is not unexpected. The linear fit of the distribution suggests that the overestimation results from a large bias by the MT. Therefore, we perform a new comparison with the reference station by applying the recursive method to correct the MT measurement. The corrected relative humidity aligns better with the reference values: the tendency remains almost unaltered from the original measurement and the bias correction is sufficient to provide a more reasonable agreement. As for the intra-sensor variability test, truncation at the second iteration is sufficient to obtain a reasonable performance of the MT under real atmospheric conditions.

For a more quantitative evaluation of the performance of the MT, Table 5 condenses the values for the coefficient of determination and the errors computed for the air temperature, relative humidity, and the corrected relative humidity during both seasons. All three variables score a high linear correlation with the reference with a clear tendency of the MT to overestimate the reference values. Winter air temperature shows a positive percentage error, as witnessed by the measurement bias for the 67% and the random error for the 33% (computed as the ratios $\mathrm{RMSE_s}$/RMSE and $\mathrm{RMSE_u}$/RMSE, respectively). Similar behavior is found for the relative humidity, with an even larger percentage error entirely owed to its positive branch, and a large $\mathrm{RMSE_s}$ over the $\mathrm{RMSE_u}$ ($\mathrm{RMSE_s}$/RMSE$\gg$$\mathrm{RMSE_u}$/RMSE). The correction adopted for the relative humidity reduces the distribution errors, favoring a larger balance between over- and under-estimation as well as bias and random errors. The correction severely reduces the total percentage error on relative humidity, balancing positive and negative errors that on their own are smaller than the MPE for the air temperature. The correction also reduces the error due to the bias to a value that is smaller than the random error, with the $\mathrm{RMSE_s}$ being caused by the remaining overestimation tendency at small relative humidities (see Fig. 13b). During summer, the MPE quantifies the air temperature overestimation at a 2.9% of the measurement, with an overall reduction of the percentage error compared to the winter period. Once again, the discrepancy for air temperature is distributed in a bias responsible for the overestimation (accounting for 55% of the error) and in the random error responsible for the



**Table 5.** Coefficient of determination $R^2$, total (RMSE), systematic ($\mathrm{RMSE_s}$), and unsystematic ($\mathrm{RMSE_u}$) root mean square errors, and total (MPE), positive ($\mathrm{MPE_P}$), and negative ($\mathrm{MPE_N}$) mean percentage errors computed between the measurements of the MT used as the predictor ($x$) and those of the weather station used as a reference ($y$).

| | Winter | | | Summer | | |
| | $T$ | $RH$ | $RH_c$ | $T$ | $RH$ | $RH_c$ |
|---|---|---|---|---|---|---|
| MPE | 4.9% | 9.3% | 0.4% | 2.9% | 19.8% | 0.6% |
| $\mathrm{MPE_P}$ | 5.0% | 9.4% | 3.3% | 3.5% | 19.8% | 7.3% |
| $\mathrm{MPE_N}$ | -1.9% | -0.9% | -2.8% | -1.6% | 0% | -4.9% |
| RMSE | 1.1°C | 9.4% | 4.1% | 3.6°C | 18.8% | 7.0% |
| $\mathrm{RMSE_u}$ | 0.4°C | 1.6% | 2.2% | 1.5°C | 10.4% | 4.6% |
| $\mathrm{RMSE_s}$ | 0.7°C | 7.7% | 1.9% | 2.0°C | 8.4% | 2.4% |
| $R^2$ | 0.96 | 0.97 | 0.95 | 0.95 | 0.88 | 0.88 |

RMSE $= \left(\frac{1}{n}\sum_{i=1}^{n}(x_i - y_i)^2\right)^{1/2}$, MPE $= \frac{100}{n}\sum_{i=1}^{n}\frac{x_i - y_i}{x_i}$, with $x_i$ the iteration of variable $x$ from the MT, $y_i$ that for the reference weather station, $n$ the number of finite data points in $x$. $\mathrm{MPE_P}$ and $\mathrm{MPE_N}$ are computed similarly to MPE, but summing only the positive and negative values of MPE, respectively

distribution spread around the reference value (45%). The linearity of the fit is respected. The relative humidity measured with the MT overestimates the reference by 20% on average (see MPE in Table 5, being entirely outside the instrumental error (Fig. 13d). This causes the MT to observe multiple saturation conditions while the reference relative humidity rarely exceeds 90%. The Stevenson screen and the reduced ventilation inside it can favor stagnant conditions thus increasing the relative humidity. However, the variability observed in the measure of the relative humidity during the intra-sensor variability test hinders this

assumption. The RMSE and its components describe a discrepancy from the reference which is larger than the winter period, and mostly due to the unsystematic error. This is caused by the large amounts of values at saturation the MT detects while the reference station observes relative humidities in the range 70–95%. In addition to the distribution spread, a bias is also evident but the linearity of the fit is respected. With the usual 2 iterations of the correction method, we observe a huge improvement in the agreement between MT and weather station data. Relative humidities below 50% seem unaltered by the correction. Small

values of relative humidity linked with large values of air temperature imply that humidex is determined mostly by the second quantity and less so by the first (see Eq. 7). Since the MT temperature does not change in the recursive method, $\delta$HDX and $\delta T_d$ are small and $T_d^c \to T_d$ as well as $RH_c \to RH$. This shortcoming of the correction did not occur during the intra-sensor variability test as the maximum air temperature was 8°C smaller than this investigation in Genova.

The recursive method enables us to further discuss the dew point and humidex index. Neither quantity is measured by the

weather station, but they can be retrieved using known empirical formulations and then compared with those retrieved directly from the MT and computed through the iterations. Specifically, the dew point $T_d$ for the weather station is derived according





to the Magnus formula, so that

$$T_d = \frac{c_1 \left( ln \left( \frac{RH}{100} \right) + \frac{c_2 T}{c_1 + T} \right)}{c_2 - ln \left( \frac{RH}{100} \right) - \frac{c_1 T}{c_2 + T}}, \tag{16}$$

where $c_1$=243.04°C and $c_2$=17.625°C according to Alduchov and Eskridge (1996). This formulation is consistent with a
relative humidity resulting from the ratio of water-vapor partial pressure and its value at saturation, and in line with other
formulations based on the Clausius Clapeyron equation (Lawrence, 2005). For consistency, the humidex from the weather
station is computed by using a Magnus formulation as well according to Eq. 7.

Figure 14 displays the distributions of the MT (directly derived by the sensor), weather station (Eqs. 16 and 7), and iterated
values of $T_d$ and HDX. The distributions of MT data and that retrieved using the weather station share similar shapes but with

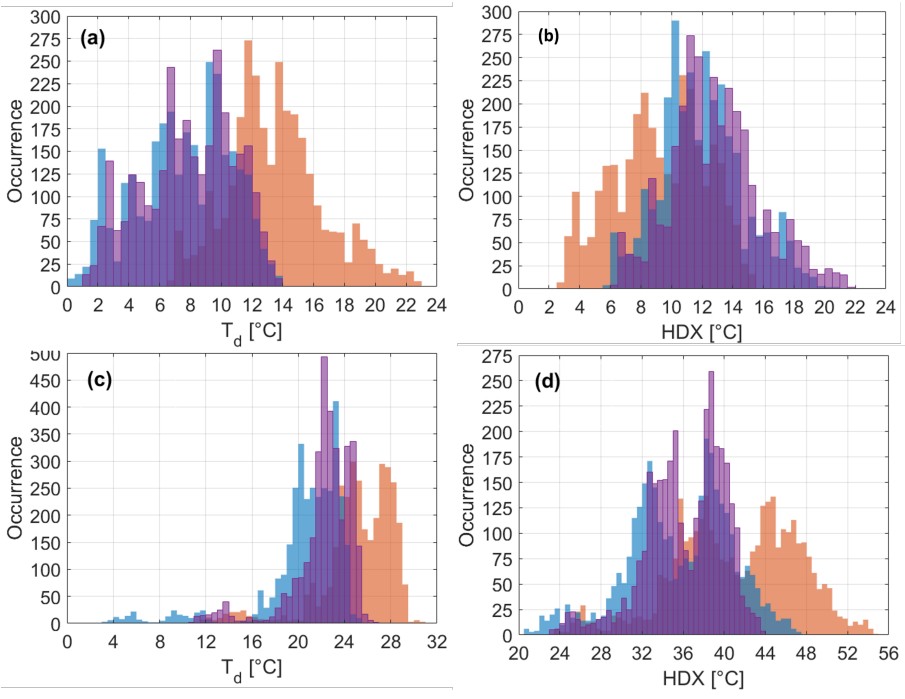

**Figure 14.** Dew point $T_d$ (a) and humidex HDX (b) distribution comparisons between the variables directly computed by the MT (red), the variables derived from the weather station data using Eq. 16 for $T_d$ and Eq. 7 for HDX (blue), and the variables obtained after 2 iterations of the recursive method seen in Sect. 4.2 (purple). Panels (a) and (b) refer to the winter period, (c) and (d) to the summertime. The bin width is 0.5°C on an equal number of bins among the distributions

a bias shifting one from the other. The dew point from the MT overestimates the distribution from the weather station, while
an underestimation is observed in the humidex. This turnover is due to the formulations adopted for $T_d$ and HDX, with the
first being largely dependent on $RH$ (where the MT largely overestimates the weather station) and the second more on $T$.
During summer only, the distributions are shaped similarly with the dew point resembling a skewed normal distribution with





a longer left tail while the humidex is a bimodal distribution covering the whole range of the index. The recursive correction

method adopted for $RH$ rescales the distributions of $T_d$ and HDX, aligning them to the weather station's ones with seasonal discrepancies. During winter, the corrected distributions display longer tails and fewer occurrences at the peaks, but they provide a general improvement in line with the relative humidity.A During summer the corrected distributions encompass restricted ranges and more occurrences at the peaks.

The comparison obtained for these winter and summer periods accomplishes two goals of this whole investigation:

– The recursive method to correct the relative humidity is proven resourceful in retrieving the true relative humidity of the ambient air, along with a reasonable estimation of the dew point and humidex index;

– The MT can capture the thermo-hygrometrical properties of the ambient air with an outdoor accuracy that resembles the reference weather station, but overshoots the combined instrumental errors.

These accomplishments support the use of the MT as a fixed weather station (under customized shielding conditions) to monitor

the mean thermal and hygrometric state of the atmosphere.

### 4.4 Mobile Comparison

The final test to assess the MT's data quality is performed under the normal operational mode of the sensor. An instrumented cargo bike is used as a benchmark for air temperature and relative humidity data quality, as described in Sect. 3.4. The comparison is performed by incorporating the measurements from all 8 cycling sessions into a single dataset. The statistical char-

acteristics of each variable pair (MT and cargo bike) were computed on this single dataset. As we will argue in this section, we believe the loss of information from each session is not relevant for the quality test since there is not a clear change of behavior in their data distribution. Only the coefficient of determination is truly overestimated considering the single dataset; for this reason, it is not computed.

Figure 15 shows the air temperature measured by the MT as a function of that from the cargo bike. The temperature range of

each session is contained within 1°C despite the cycling path covering both the city center and surroundings of Wageningen. Several ambient factors can be responsible for this small thermal excursion, but they are beyond the scope of this investigation. The key information is that the MT captures the same thermal excursions within the entire temperature range collected during the cycling sessions. Moreover, the majority of the temperature data falls into the error range across the bisector of the comparison, entailing a better quality of data distribution compared to the results for the fixed-location comparison (see Sect. 4.3). A

small tendency to overestimate the air temperature from the cargo bike can be observed for each cycling section (Fig. 15). The linear fit displays this overestimation, suggesting the presence of an average bias of 0.3–0.5°C in the MT data (which remains within the instrumental error range of the MT and cargo bike sensor combined).

The relative humidity measured by the MT shows the well-known overestimation problems we have observed during the previous tests (Fig. 16a). Relative humidity from the MT is consistently outside the error range across the bisector in compar-

ison with the cargo bike, and the overestimation increases with decreasing relative humidity. This behavior is observed also during the winter period of the fixed-location comparison, where the increasing overestimation of the reference-station data at





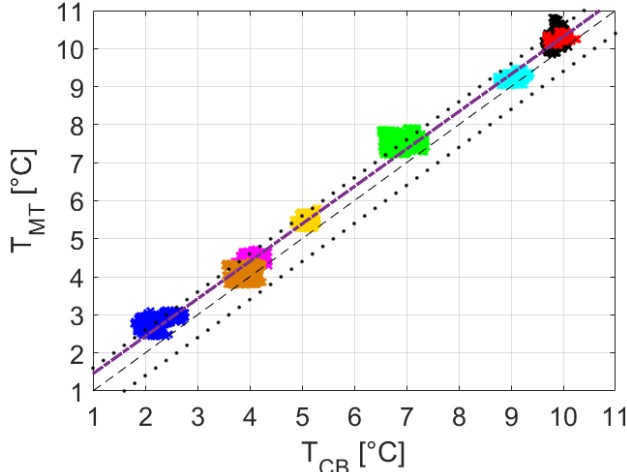

**Figure 15.** Air temperature $T$ comparison between MT (subscript MT) and cargo-bike (subscript CB) measurements including all the cycling sessions each of them identified by a separate color. The dashed purple line is the linear best fit. The dashed line is the bisector, the dotted lines identify the range given by the sum of the MT and cargo-bike instrumental errors around the bisector, that is $\pm 0.6$°C

smaller relative humidity was associated with an increasing overestimation at larger air temperatures (see Fig 13a). Under fair weather, the typical behavior for air temperature is to increase when relative humidity decreases, and vice versa. Therefore, an overestimation at a large temperature would call for an overestimation at a small relative humidity as observed. However, the

result of the cargo-bike test shows that air temperature and relative humidity are completely disjoint and that the performance of the temperature sensor is once again far better than the relative humidity one. Applying the recursive method, the corrected relative humidity improves the agreement with the cargo bike data. Both overestimation and the increasing discrepancy at smaller values are partially corrected, with a general improvement in the MT data quality. Most of the corrected values of relative humidity fall into the error range, and the fit becomes almost parallel to the bisector.

As for the fixed-location comparison, the MPE (total, positive, and negative branches) and the RMSE (total, systematic, and unsystematic) are computed and their values are listed in Table 6. Air temperature shows MPEs and RMSEs in line with the winter period of the fixed-location test (see Table 5). The errors fairly describe the overestimation trend ($\mathrm{MPE} > \mathrm{MPE_P} \gg \mathrm{MPE_N}$) as a result of the MT positive bias in the air temperature ($\mathrm{RMSE_s} \gg \mathrm{RMSE_u}$). The improvement introduced with the recursive method (after 2 iterations) is quantified by the error differences between $RH$ and $RH_c$. The overestimation problem

of the MT remains even after the correction application, but its amplitude is drastically reduced. The recursive method decreases the MPE (both positive and total) by 73% and the RMSE (total and systematic) by 86%. The positive bias between the MT and the cargo bike remains as the slight increase in the random error fraction $\mathrm{RMSE_u}/\mathrm{RMSE}$ is not sufficient to counterbalance the relation $\mathrm{RMSE} \geq \mathrm{RMSE_s} \gg \mathrm{RMSE_u}$ observed for the measured $RH$. As for the air temperature, the corrected relative humidity has statistical errors in line with the fixed-location test (worse than winter, better than summer periods).

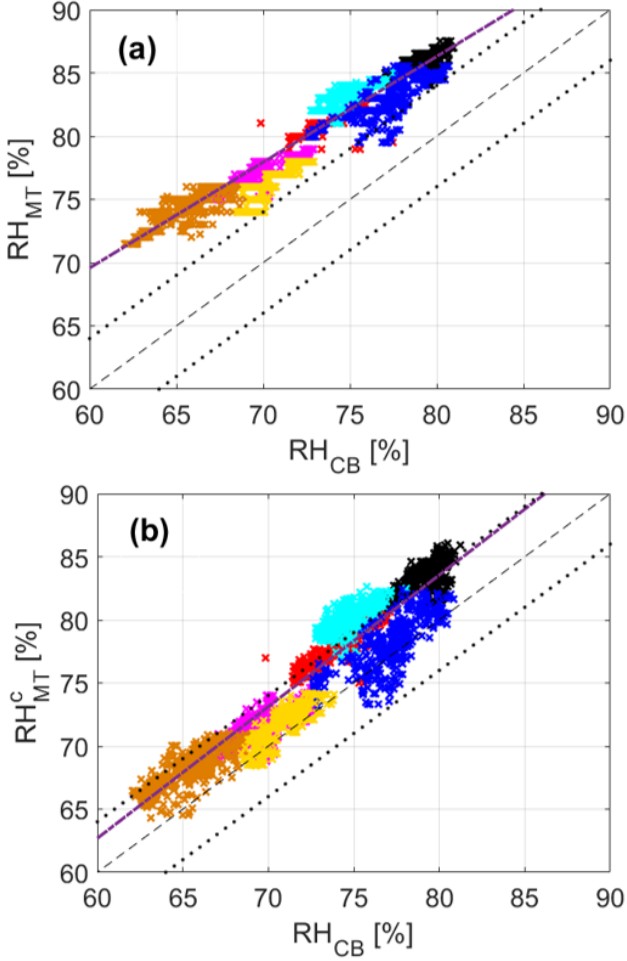

**Figure 16.** Measured $RH$ and corrected $RH^c$ relative humidity comparison between MT (subscript MT) and cargo-bike (subscript CB) measurements including all the cycling sessions each of them identified by a separate color. The dashed purple line is the linear best fit. The dashed line is the bisector, the dotted lines identify the range given by the sum of the MT and cargo-bike instrumental errors around the bisector, that is ±4%

To further investigate the reasons for the overestimation, we inspect the incoming solar radiation and ventilation speed measured by the cargo bike. Being unshielded and unventilated, the MT can increase the temperature of its case or that of the air sampled when stagnant. Both conditions can lead to an overestimation of the air temperature and uncertain implications for the relative humidity, despite the software correction of the RECS. On one hand, the increasing temperature of the sensor case can dry the air sample, reducing the relative humidity. On the contrary, stagnation can lead to saturation and an increase

in relative humidity. To investigate the impact of both quantities on the sensor performance, the percentage deviation of air



**Table 6.** Total (RMSE), systematic ($\mathrm{RMSE_s}$), and unsystematic ($\mathrm{RMSE_u}$) root mean square errors, total (MPE), positive ($\mathrm{MPE_P}$), and negative ($\mathrm{MPE_N}$) mean percentage errors, and mean absolute error (MAE) computed between the measurements of the MT used as the predictor ($x$) and those of the weather station used as a reference ($y$). Values are computed including all cycling sessions

|  | $T$ | $RH$ | $RH_c$ |
|---|---|---|---|
| MPE | 6.4% | 11.5% | 3.0% |
| $\mathrm{MPE_P}$ | 7.1% | 11.5% | 3.3% |
| $\mathrm{MPE_N}$ | -2.3% | 0% | -1.2% |
| RMSE | 0.6°C | 11.9% | 5.4% |
| $\mathrm{RMSE_u}$ | 0.2°C | 1.7% | 1.9% |
| $\mathrm{RMSE_s}$ | 0.4°C | 10.2% | 3.5% |
| MAE | 0.33°C | 7.65% | 2.58% |

RMSE $= \left(\frac{1}{n}\sum_{i=1}^{n}(x_i - y_i)^2\right)^{1/2}$, MPE $= \frac{100}{n}\sum_{i=1}^{n}\frac{x_i - y_i}{x_i}$, MAE $= \sum_{i=1}^{n}|x_i - y_i|$ with $x_i$ the iteration of variable $x$ from the MT, $y_i$ that for the reference weather station, $n$ the number of finite data points in $x$. $\mathrm{MPE_P}$ and $\mathrm{MPE_N}$ are computed similarly to MPE, but summing only the positive and negative values of MPE, respectively

temperature and corrected relative humidity are computed as

$$\Delta_\% = 100\frac{\chi_{MT} - \chi_{CB}}{\chi_{MT}}, \tag{17}$$

where $\chi$ is either $T$ or $RH^c$ and the subscripts refer to the MT (MT) or the cargo bike (CB). An average among bins of incoming solar radiation serves as a bin-driven MPE (named bin-averaged percentage deviation) to compare with those in
Table 6. Specifically, the MPE is computed within bins of 50 Wm$^{-2}$, averaging the $\Delta_\%$ for each variable. Figure 17 displays the percentage deviations as a function of the incoming solar radiation and the ventilation speed. As a common feature, air temperature and relative humidity deviations cover a large range of variability. Temperature distribution displays a positive trend, increasing the overestimation of the MT with increasing radiation. For solar radiation in the range 0–200 Wm$^{-2}$, the bin-averaged percentage deviation is close to the MPE retrieved for the whole distribution, describing a positive bias below the
8% compared to the cargo bike. At the radiation peak of the investigated period, the bin-averaged percentage error is almost 15%, doubling the low-radiation condition. On the contrary, relative humidity has a negative trend decreasing the MT deviation from the cargo bike with increasing radiation, reaching an overturning of the deviation sign starting at 200 Wm$^{-2}$. The bin-averaged percentage deviation is approximately 1% higher than the MPE in the range 0–200 Wm$^{-2}$, while decreasing below a value of 1% above 300 Wm$^{-2}$. The ventilation speed does not show a neat impact either on temperature or relative humidity.
At 200–250 Wm$^{-2}$ the percentage deviation of air temperature increases with the ventilation speed while decreasing within the previous and following 50-Wm$^{-2}$ bin. Once again, the relative humidity deviation behaves in opposition to air temperature.





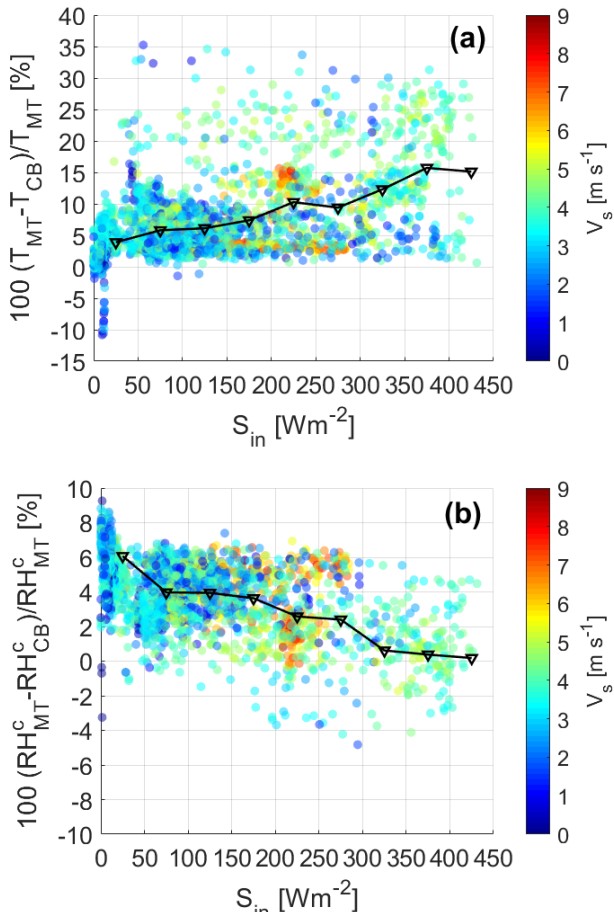

**Figure 17.** Percentage deviation of air temperature $T$ (a) and corrected relative humidity $RH^c$ (b) as a function of the incoming shortwave radiation $S_{in}$ and the ventilation speed $V_s$, including all the cycling sessions. The black line with diamonds is the MPE computed on each 50-Wm$^{-2}$ bins of incoming shortwave radiation and plotted in the middle of the bin

The comparison with the measurements from the cargo bike has proven that:

– The recursive method is a resourceful tool for retrieving the true relative humidity of the ambient air under the standard working condition of the MT;

– The MT tendency to overestimate both air temperature and relative humidity (mostly within the instrumental error) is a shortcoming of the sensor not induced by external factors (e.g., non-standard operational conditions);

– The MT can capture the thermo-hygrometrical properties of the ambient air while operating on the move with an accuracy that resembles a certified mobile weather station, but overshoots the combined instrumental errors.





These accomplishments support the use of the MT as a mobile weather station to monitor the mean thermal and hygrometric
state of the atmosphere.

## 5  Discussion

### 5.1  Notes on the recursive method

The recursive method is the correction introduced to modify the value of the relative humidity through an iterative correction of
the dew point and the humidex. The underlying hypothesis of this method is that the modification imposed on the dew point is
proportional to those on humidex and relative humidity, with air temperature being constant. This hypothesis is most likely true
as soon as the modification is small since relative humidity can oscillate around a constant air temperature. The performance of
the MT temperature sensor also suggests that large modifications can be possible, those being related to a bad MT performance
in measuring relative humidity rather than real atmospheric variability. However, no physical principles support this method,
but rather a tentative solution to adjust a biased sensor.

Being based on the error propagation theory, the recursive method is divergent by definition. Both corrective terms $\delta T_d$ and
$\delta$HDX are positive at each iteration, rapidly bringing $RH_c$ to unrealistic values. The introduction of a negative sign in $\delta T_d$
equation alternates positive to negative contributions to $T_c$; nonetheless, Fig. 18 shows a slow but continuous decrease of the
$RH_c$ with a divergent ending after several iterations. Note that the first iteration is only needed to reproduce the measurements,

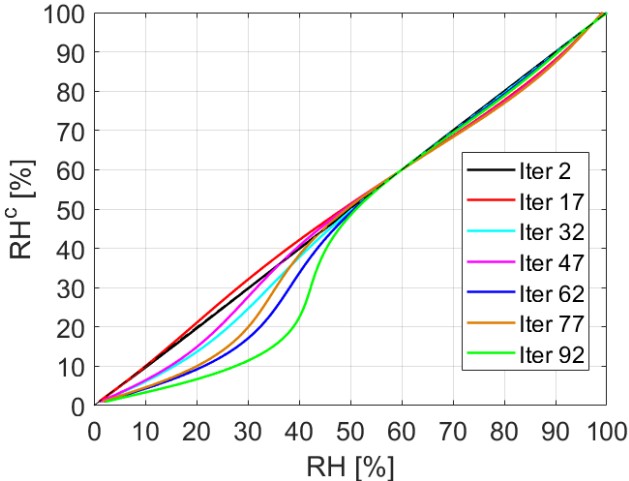

**Figure 18.** Corrected relative humidity $RH_c$ as a function of the measured one $RH$ at different iterations. The recursive method is built
among the full $RH$ range and for air temperatures between -5°C and 45°C. The dashed line is the bisector

so we can argue that the true iteration is $i-1$. The number of iterations required to reach a non-physical value of $RH_c$ depends
on the reciprocal variation of air temperature and relative humidity. In the specific case of Fig. 18, as $RH$ increases from 0% to
100%, $T$ decreases from 45°C to -5°C, covering a full range of realistic values for mid-latitudes. This implies a perfect scenario





where air temperature and relative humidity are perfectly (and inversely) correlated, while reality mostly shows oscillations of air temperature around each value of relative humidity and vice versa. A more complex variation of air temperature with relative humidity modifies the effect of the recursive method, amplifying or decreasing its impact at different values of $RH$.

Nonetheless, the scope of this section is to argue on the divergent nature of the recursive method. Breaking the recursive method after a few iterations is therefore fundamental to avoid the result divergence and to ensure the final $RH_c$ is "close" to the original $RH$. In a way, a small number of iterations constrains $RH_c$ to $RH$. For the tests conducted in this manuscript, the recursive method only necessitated 2 iterations, involving a small decrease of relative humidity compared to the original measurement. Although the number of tests is limited, the constant number of iterations used to reach the best agreement with

the reference can be a recipe for future applications.

## 5.2 Performance comparison with other mobile sensors

The world of mobile sensors and stations comprises many suites, ranging from research-grade instruments mounted on moving carriers to low-cost sensors specifically designed for mobile sensing. This comes with a variety of nominal (i.e., evaluated by the manufacturer through laboratory testing under ideal measuring conditions) accuracy that depends on the quality of the

sensor's suite. From the literature survey of this paper, the accuracy ranges between ±0.1°C to ±1°C for air temperature and ±1.5% to ±5% for relative humidity, with the MT falling into the ranges. Under outdoor ambient conditions, operating a sensor with a non-optimal setup (e.g., a sensor designed for monitoring in a fixed location used as a mobile device) or using a low-cost sensor decreases the chances of meeting the manufacturer accuracy due to the response time of the instrument to the non-stationarity of the ambient air. In this section, we contextualize the performance of the MT in comparison with its

reference to other validation studies for mobile sensors from the literature. Although aware of the limits of compatibility of the different mobile sensors and comparability of the collected data (Schering et al., 2022), this comparison serves as a qualitative way to rank the performance of MTs with their competitors. We will take the results obtained for the mobile comparison (Table 6 in Sect. 3.4) as the sensor was used under normal operational mode. Considering sensors usually mounted in fixed stations, HOBOs (Onset Computer Corporation, USA) are the most widely used for mobile sensing due to their compact

design, moderate cost, and research-grade reliability. Among those, Qi et al. (2022) adopted a shielded HOBO MX2302 (air temperature accuracy of ±0.2°C) mounted on a cart at 1.5 m agl. Along the measuring track, the cart passed 14 fixed stations in the area used as a reference. The comparison showed that 80% of measurements laid into the accuracy range of the instrument, and the MAE after reaching the optimal measuring setup was in the range of 0.14°C to 0.62°C in line with this paper. Tsin et al. (2016) used a shielded temperature sensor (Met One 064-2, with an accuracy of ±0.1°C sampling at 10 s) mounted on a pipe;

walking sampling; comparison with fix station nearby revealed an underestimation in the range of 2°C and a poor agreement with data from Landsat (with a coefficient of determination $R^2 = 0.04$–0.38). Crowdsourcing air temperature using Netatmo urban weather stations (https://www.netatmo.com), Meier et al. (2017) obtained a RMSE in the range 0.5–1.5°C for spatially aggregated raw data for hourly and daily urban air temperatures. A better performance was obtained by Liu et al. (2017) who opted for a shielded HOBO-THB-M002 (air temperature accuracy of ±0.2°C) mounted on a bicycle, obtaining a RMSE and

MAE within 0.3°C when comparing the local UHI intensity obtained from mobile sensors and that from 9 reference stations



along the track. The bicycle is among the top choices for mobile sensing, combining a good spatial coverage of the investigated domain and a low moving speed facilitating the comparison with the fixed reference. The bicycle is also a practical solution for low-cost sensors owing to the low-cost philosophy (no emission, low if no energy consumption, small price of the vehicle). Rodríguez et al. (2020) developed and tested a self-made temperature sensor obtaining a deviation from the fixed reference

along the path in line with the sensor accuracy ($\pm0.5$°C). Worse results were obtained by Vieijra et al. (2023) approaching the accuracy ($\pm1$°C) of its air temperature device based on the Adafruit BME280 sensor (developed by Bosch) when compared with fixed references.

     Other authors preferred to derive the real accuracy of the mobile sensor in isolated chambers before using it in the field. Although close to ideal conditions, this test offers a second accuracy computation independent from the manufacturer. Selected

chambers for the scope are calibration ovens (Skoulika et al., 2014) and isolated chambers (Cao et al., 2020), where variations in temperature and humidity are controlled. In the study from Skoulika et al. (2014) the real accuracy of the BC15 thermo-hygrometer sensor (TROTEC International GmbH & C. S.a.s.) was estimated in the range of $\pm0.2$°C to $\pm1$°C, while Cao et al. (2020) reaches a maximum real accuracy of $\pm0.48$°C and $\pm1.9$% for the sensor suite unit Smart-T (developed at Yale University). Although we cannot perform a similar experiment, we can have an evaluation based on the field experiment,

knowing that we were exposed to a higher and faster ambient variability than the laboratory tests. Being a mean approximation of data distribution, the linear fit for air temperature and relative humidity from the mobile test can be used to infer the outdoor accuracy of the MT. We can argue that the outdoor accuracy for air temperature is in the range $\pm0.3$°C to $\pm0.5$°C while the corrected relative humidity ranges between $\pm3$% and $\pm5$%, both values in line with the ranges depicted in the from the literature.

Finally, Cecilia and Peng (2022) performed a first validation test of the MT, monitoring the air temperature and relative humidity against two 5400 WBGT Heat Stress Tracker (Kestrel Instruments) along a bicycle path traveled at low speed. The comparison showed a large agreement between the MT with its references, with a coefficient of determination for both variables $R^2$=0.74 compared to $R^2$=0.64 obtained as an average of all sessions performed within the carbo-bike experiment of this paper.

## 6   Conclusions

This paper explores the outdoor performance of a novel mobile sensor in performing consistent measurements of atmospheric microclimate characteristics. The sensor under investigation is the MeteoTracker, a mini-weather station suitable for monitoring the thermo-hygrometric environment on the move. Three validation tests have been performed in three different urban contexts to evaluate the sensor precision and accuracy compared to the reference, under different climates and seasons. Specifically, the city center of Bologna hosted an intra-sensor variability test to ensure the consistency of measurements from different

MTs under similar ambient conditions. In Genova, a comparison with a reference meteorological station has been performed using the MT as a fixed monitoring station. Finally, a comparison with reference sensors has been carried out in Wageningen under normal operation mode, thanks to the instrumented cargo bike developed herein. Several statistical parameters have been adopted to quantify the MT performance in all tests, and compared with similar sensors from the literature. Satisfactory





performance was reached by the MT when the RMSE of each test was close to the instrumental errors of the involved sensors,
which is sufficient for the scopes the sensor was built for, i.e., crowdsourcing monitoring. The results showed that under
optimal operational use (on the move), only the air temperature satisfied the required performance with high-quality statistical
scores, that is RMSE smaller than the sensor accuracy and MPE $\leq 6.4\%$ due to a positive bias. The relative humidity was
the worst performing variable with large intra-sensor variability (MAPE up to 32%) and discrepancy from the reference (with
similar RMSE and MPE between 9% and 20% according to the test). The other measured variables or derived quantities were
affected by large statistical errors when dependent on the relative humidity, while pressure and altitude scored satisfactorily
but improvement could be made.

Data-driven corrections were derived from known analytical formulations and online services for the bulk atmosphere and
revealed mandatory for a quality improvement of atmospheric pressure, location altitude, relative humidity, dew point, and
humidex index. Altitude was recomputed at each GPS location using a web server and the result corrects the atmospheric
pressure through the psicrometric formula. Dew point, humidex, and relative humidity were involved in a recursive method
based on bulk formulations of all three quantities that consistently improved after two iterations regardless of the experimental
type conducted in this investigation. Despite the corrections, the general trend of the MT is to overestimate the reference sensor
more often by a constant factor (bias) and less so by random uncertainty. The validation tests also revealed the role of ancillary
variables on air temperature and relative humidity. Solar radiation proves to have contrasting impacts on the MT performance:
an increase in solar radiation intensity enhances the gap between the air temperature measured with the MT and the cargo bike,
decreasing the performance of the first sensor; on the contrary, relative humidity reduces the sensor's gap with increasing solar
radiation, enhancing the performance of the MT.

The outdoor performance of the MT aligns with most of the mobile sensors whose accuracy and precision were evaluated
in the field. Its real outdoor accuracy falls in the range found in the literature for similar mobile sensors, and it is observed
between $\pm 0.3°C$ and $\pm 0.5°C$ for air temperature and between $\pm 3\%$ and $\pm 5\%$ for relative humidity. However, this partial and
very qualitative ranking does not account for the disparity in the cost of the sensors nor the different experimental designs
used to assess the outdoor their real accuracy. Considering the trademark between the cost and the quality of the sensor, the
MT marks itself as a valid solution from crowdsourcing experiments to long-term route monitoring, providing quality data for
atmospheric monitoring.

*Data availability.* Data from the MeteoTrackers are available for visualization on the customized platform at https://app.meteotracker.com/
under the authors name "bologna_living_lab_*", "genova_living_lab_*" and "amsterdam_living_lab_*", where * is a one-to-two
digit number representing the MT ID in the location. Reference data can be made available upon request.



*Author contributions.* FB developed the theoretical framework and performed the data analysis. Support for the data analysis was given according to the experimental site by EB, CC, AGN, MM, EP, SP and GS. All authors contributed to the test designs and implementation.
SDS, AP, and GS supervised the research approach

*Competing interests.* The authors declare they have no conflict of interest.

*Disclaimer.* This paper reflects the author's views. The European Commission is not responsible for any use that may be made of the information it contains.

*Acknowledgements.* This research has been supported by the project I-CHANGE (Individual Change of HAbits Needed for Green European
transition, https://ichange-project.eu/). The project has received funding from the European Union's Horizon 2020 research and innovation program under grant agreement No 101037193. This study uses data generated by the Ruisdael Observatory, a scientific research infrastructure that is (partly) financed by the Dutch Research Council (NWO, grant number 184.034.015). The authors sincerely acknowledge IoTopon Srl, the manufacturer of the MT for technical support for the field testing and for having shared their methods for computing the derived quantities. We also thank Maryam Sarfraz for her support on data acquisition in Bologna, Vincenzo Mazzarella for technical collaboration
with the data analysis in Genova, and Henk Snellen for his technical support on the observations done in Wageningen.



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
