# Peer review of "Performance Evaluation of MeteoTracker Mobile Sensor for Outdoor Applications"

_Atmospheric Measurement Techniques, 2023_

## Author Response (AR1)

Dear Editor, Dear Reviewers

Thanks for having read and corrected our manuscript. In this document, we have reported in italics your comments posted in the discussion area for our manuscript, divided per questions and topics and one-by-one responses from our side are provided in regular font. Where we believe they were already exhaustive, comments responses are taken directly from our posts in the discussion area. Major comments and modifications to the manuscript refer to: (i) device memory overload (RC1, comment 5), but we believe this is a contained risk since the memory usage of the connected smartphone is very small; (ii) measure of temperature modified by the vicinity to car body (RC2, comment 11), where we agree to the reviewer but all the measure to minimise this issue were taken and explained. References to changes in the manuscript are also listed and they refer to the revised version of the manuscript with track-change. Modifications of the manuscript are also reported in this document with additional text being in blue and barred text when deleted. Two versions of the revised manuscript are submitted: a first one with modifications in track-change and a "polished" one where the manuscript is presented with modifications implemented.

Thank you,

**RC1: Anonymous Referee #1**

*1) The proposed contribution presents a detailed investigation of the performance of a commercial device aimed at providing high spatial resolution for microclimate investigation through mobile monitoring, in an IoT philosophy. The device is accessible via smartphone and data can be easily shared among the community on a dedicated platform according to the user available license. The topic is of great relevance given the rising awareness on the implications of complex urban morphology and ecosystem on intra-urban microclimate variability which impacts, among others, urban communities' resilience. The authors comprehensively explore the reliability of the commercial device object of the study through three experimental steps that provide a clear view of the instrument potentials and limitations. Moreover, the authors implemented a recursive method to correct the relative humidity data that have been proven to significantly increase the reliability of measurements collected through the system under the formulated hypothesis. The methodology is accurately presented and could be of reference to future studies involving the usage of novel mobile systems for microclimate investigations, that are expected to increase in number given the relevance of the topic.*

1R) We thank the referee for the comment, and we are glad for the overall positive evaluation of our manuscript. We have made an effort to further review and improve the manuscript according to the specific comments raised by this and the following reviewers.

*2) Just minor comments from my side: please check the equations numbering in the paper, it seems that the recalled equation 3 in section 4.1 is missing.*

2R) We have revised the missing referencing in section 4.1. We have also revised Equations 2 and 3 grouping them under a single label (Eq. 2) which we believe is more adequate. All references to equations in the manuscript are therefore updated.

*3) typo at page 2, line 50 "oOftentimes"*

3R) Corrected (line 50)

*4) page 3 line 93 "mwith".*

4R) Corrected (line 93)

*5) A final comment related to the usability of the system in long-term route monitoring: the lack of internal memory may be a limitation since the device needs to be constantly connected to a smartphone which is feasible, but a backup plan is always recommended. Not sure about the economical and technical implications due to the integration of a minimum internal memory capacity.*

5R) We have not explored long-term route monitoring but a few isolated long trips have been made and we have not observed any major limitation either with the sensor battery (it should last for 250 hours according to the manufacturer) nor with the storage memory. Indeed, data are not stored in the sensor but flow directly into the smartphone application so the limitation is given by the available memory on the smartphone. Nonetheless, this problem is quite limited as the application occupies approximately 50 MB and each data point is <2 kB (which means that for a 500 km track, MT data occupies <17 MB). We have added this information to the manuscript, it could be helpful for the general overview of the sensor. Regarding the use of the MT without a constant connection to a smartphone, there is a version of the sensor named stand-alone where the MT is connected to a cellular/GNSS module, but we have not tested this solution.

Modification in the manuscript:

line 123: The station has an internal memory that ensures up to 250 h of usage and it is remotely controlled using a customized application on the user's smartphone, [...]

Section 2.2.1: As previously described, the MT is not supplied with an internal memory; therefore, data collected during the monitoring sessions flow directly into the smartphone application via Bluetooth and are stored in the smartphone's internal memory. The limitation to data storage is only due to the available internal memory of the smartphone connected to the MT. Limiting these considerations to the experience gathered during the monitoring activities described within this manuscript, the amount of memory necessary for running an MT monitoring session is small: the smartphone application takes 53.1 MB of space and each data point collected within a monitoring session is less than 2 kB. As a practical example, a monitoring session of 500 km at the finer resolution generates less than 17 MB of data. Once the monitoring session is ended and data is uploaded on the platform, data stored on the smartphone can be deleted, saving internal memory.

**RC2: Anonymous Referee #2**

*6) The authors focus on evaluating the metrological performance of a commercial device designed for monitoring microclimate in urban areas with high spatial resolution, compliant with Internet of Things (IoT) principles and developed with low-cost sensors. This device can be connected to a smartphone to record and share data on a dedicated platform. The authors conduct three experimental phases to thoroughly evaluate the reliability of the device and introduce a recursive method to improve the accuracy of the humidity data. The methodology is well presented.*

6R) We thank the reviewer for the revision and the comments provided. We are glad to read the general appreciation of our research work. An overall revision of the manuscript has been carried out to improve it, correct the typos and references in the text, and add the units of measurement that were missing. Responses to the specific comments are listed below.

*7) line 50: please revise the typos*

7R) Done (line 50)

*8) Table 1: Please provide the unit of measurement for each variable*

8R) Units of measurement are added in the "Variable" column of Table 1

**Table 1.** Significant variables measured by the MT, with accuracy and operational range according to the manufacturer.

| Variable | Accuracy | Operational range |
|---|---|---|
| Air temperature $T$ [°C] | $\pm 0.5$°C under solar radiation and $V_S > 7 kmh^{-1}$ | $-40$°C $- +125$°C |
| Relative humidity $RH$ [%] | $\pm 2\%$ | $0\% - 100\%$ |
| Atmospheric pressure $P$ [hPa] | $\pm 3\ Pa$ (relative) or $\pm 50\ Pa$ (absolute) | $-$ |
| Altitude above mean sea level $Z$ [m] | $\pm 10\ m$ (for the initial altitude value only) | $-$ |

[1]RECS, patent of the manufacturer

*9) Line 135: please explain why you did not consider internal memory*

9R) this specific sensor does not have an internal memory and this is a choice made by the manufacturer; recently (since 2023) a new version of this sensor was released with an external memory that can be attached to the sensor without the need for a smartphone (it is called MeteoTracker stand-alone) but the sale of this version was opened right after all the monitoring activities were completed so we decided to continue with our sensors. In the revised version of the manuscript, we specify that this is a choice made by the manufacturer.

Modification in the manuscript (line 136): [...] (as a choice of the manufacturer, the station does not have an internal memory). See also the new section 2.2.1 introduced also accounting for comment 5 of RC1.

*10) Line 153: it might be useful for the reader to also read the Koppen classification for the three areas considered for the test.*

10R) Thank you for the suggestion, we have added a reference to the Koppen classification for each of the three sites.

Modification in the manuscript:

lines 163-166: This choice ensures testing the sensors in different climate areas, classified following the Koppen-Geiger climate classification (Beck et al., 2018) in humid subtropical climate (Cfa), hot summer Mediterranean climate (Csa) and temperate ocean climate/subtropical highland climate (Cfb) respectively for Bologna, Savona and Wageningen[1].

footnote page 7: Following the Koppen-Geiger climate classification: C=temperate, f=no dry season, s=dry summer, a=hot summer, b=warm summer

*11) Line 191: if you also look at the data in Table 2, it is clear that there is a summer period when the three test sections were performed. So you wrote that the MTs were placed on the top of the vehicle, as close as possible to the car axis. But what about the factor of proximity to the car body (which can heat up a lot when exposed to sunlight in summer).*

11R) Indeed the air in the immediate surroundings of the car can be warmer than the air at the same height. Unfortunately, this effect is almost impossible to remove but we have tried to minimize it by keeping the

sensors on the car top away from the major heat sources of the car (engine, brakes, wheels) and where the aerodynamic effect is larger. Also putting the sensors in the front part of the car rooftop maximizes the amount of fresh air that has not interacted with the car before being sampled by the sensor. We integrated the discussion with these considerations as a best practice for the installation. As a last consideration, for the specific intercomparison test the heating effect of the car is not an important fact as soon as all sensors are impacted the same way.

Modification in the manuscript (lines 203-205): The location on the car top minimizes the sensor's exposition to the direct heat sources of the car (engine, brakes, wheels) while maximizing the exposition to the fresh air that has not interacted with the car before being sampled by the sensor.

*12) Lines 201-207: this section can be moved before section 3.3, which already refers to the comparison with fixed location*

12R) We thank the reviewer for the suggestion but we believe that this paragraph should remain in section 3.3 as it further introduces the need for a comparison with research-grade instrumentation and the difficulties that arise with a non-conventional device like the MT.

*13) Line 207: please check the reference in round brackets: maybe it is sect. 4.4 ?*

13R) Corrected (line 221)

*14) Figure 6d: Could you add the U.M.?*

14R) This quantity is dimensionless; we specified it in the manuscript (line 296)

*15) In Figure 11, the bin width does not seem to be 0.5°C, please check*

15R) Thank you, we have corrected the indication on bin width in the figure caption

16) In the caption of Figure 4, please link the suffix "c" , "d" in the headings of the table with the reference to "measured", "derived" or "corrected".

16R) Probably the reviewer was suggesting this procedure for Figure 6. Nonetheless, we believe specifying which variable is measured, derived or corrected in each figure containing data plots will help the reader, so we have implemented this correction.

Modification to figures 7, 8, 10, 12